# WHAT IF TSF: A MULTIMODAL BENCHMARK FOR CONDITIONAL TIME SERIES FORECASTING WITH PLAUSIBLE SCENARIOS

## ABSTRACT

Time series forecasting has long been constrained by history-bound, unimodal methods and benchmarks that fail to capture predictive, forward-looking context. Recent progress in large language models and multimodal alignment suggests richer possibilities, yet most existing multimodal benchmarks rely on textual descriptions that merely repeat historical patterns and can introduce misleading signals due to irrelevant context. To advance research in this area, we introduce "*What If TSF (WIT)*", a benchmark constructed around expert-crafted what-if scenarios and explicit future events. WIT encourages models not only to match historical patterns but also to reason under uncertainty, evaluating their ability to integrate multimodal signals, anticipate plausible futures, and enable conditional forecasts. By moving beyond historical pattern extraction, WIT establishes a principled testbed for scenario-guided multimodal forecasting.

## 1 INTRODUCTION

Anticipating what lies ahead is a defining ability of intelligent systems, natural or artificial. Brains and algorithms alike depend on projecting the future in order to plan, adapt, and survive (LeCun, 2022; Nayebi et al., 2023). Forecasting plays a critical role across society: businesses estimate consumer demand to guide investment, governments predict economic or energy indicators to shape policy (Goodwin et al., 2023; Coroneo, 2025), and fields from climate science (Kent et al., 2025) to epidemiology (George et al., 2019) use forecasting to transform past observations into actionable foresight.

Most forecasting methods, whether statistical or learning-based, have conventionally focused on numerical time series alone. Recent Time Series Foundation Models (TSFMs) extend this paradigm by scaling up model size and data coverage. Yet, they still primarily extrapolate historical patterns, so their advantages over conventional baselines remain unclear. Evidence from benchmarks remains divided: OpenTS (Qiu et al., 2024; Li et al., 2025a) and FoundTS (Li et al., 2024) show that statistical or supervised baselines can rival or surpass specialized TSFMs, while tabular foundation models such as TabPFN-v2 (Hollmann et al., 2023; 2025; Ye et al., 2025) achieve comparable performance without time-series-specific architectures (Hoo et al., 2025). Conversely, evaluations like GIFT-Eval (Aksu et al., 2024b) provide results more favorable to TSFMs. Overall, this points to limits of unimodal, history-based forecasting rather than scale or architecture.

The rapid development of large language models (LLMs) and advances in multimodal alignment have opened new opportunities for forecasting (Kong et al., 2025b). Unlike conventional models confined to numerical sequences, LLMs can process unstructured text and leverage external knowledge that purely time-series-based models cannot access. Emerging approaches, including representation-fusion methods and prompting-based strategies (Jin et al., 2023; Liu et al., 2024b; Requeima et al., 2024), illustrate the potential of natural language as an intuitive interface for incorporating side information.

However, recent evidence shows that the effectiveness of multimodal forecasting depends critically on the quality of textual context. A comprehensive study (Zhang et al., 2025b) finds that multimodal methods often fail to surpass strong unimodal baselines because most benchmarks pair time

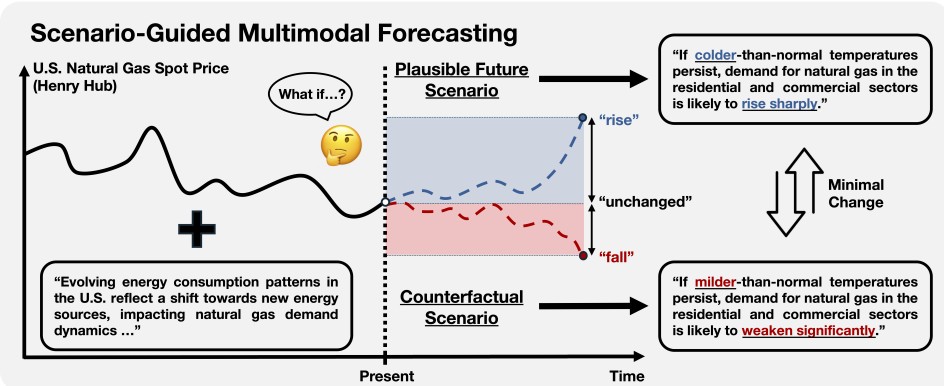

Figure 1: Overview of WIT benchmark. Our benchmark enables scenario-guided multimodal forecasting. The figure illustrates how textual information about plausible future scenarios and counterfactual scenarios can influence the directional outlook of target future time points, highlighting the role of scenario-guided context in shaping forecasts.

series with textual context that is redundant with historical numerical patterns. Retrospective narratives of past events, in particular, seldom provide genuinely predictive signals and can even hinder performance by introducing redundancy or noise.

These limitations highlight the need for multimodal benchmarks for time series forecasting that move beyond descriptive or redundant text. Instead, multimodal benchmarks should provide genuinely informative, forward-looking signals such as scenario descriptions or expected future events derived from expert knowledge. Without such signals, models may appear to benefit from multimodality while merely exploiting spurious correlations, obscuring whether they truly reason with external information. Human experts, by contrast, foresee potential contingencies through *what-if* scenarios and domain knowledge. Such integrative reasoning is indispensable in high-stakes settings where uncertainty and anticipated external shocks are not captured by time series alone. Multimodal forecasting approaches similarly seek to overcome these limitations by integrating complementary information sources beyond time series.

To address this gap, we present "*What If TSF (WIT)*", a new benchmark specifically designed to push time-series forecasting beyond historical pattern replication. WIT uniquely combines expert-crafted *what-if* scenarios and explicit future events with structured textual descriptions that encode anticipated developments and domain knowledge. This benchmark provides a well-defined foundation for directly assessing multimodal models on plausible future scenarios expressed in text, testing whether they can integrate heterogeneous signals and effectively incorporate explicit future information into forecasts. By making these capabilities measurable and comparable, WIT establishes a concrete foundation for advancing research toward multimodal forecasting methods that genuinely leverage external context and enable reasoning-driven predictions.

## 2 RELATED WORK

### 2.1 MULTIMODAL TIME SERIES DATASETS

Growing interest in applying LLMs to time series analysis has motivated the development of multimodal benchmarks that pair numerical sequences with text or other modalities. In healthcare, MIMIC (Johnson et al., 2016; 2020) has long combined physiological signals with clinical notes, while in finance, datasets linking stock prices to news and reports (Xu & Cohen, 2018; Wu et al., 2018; Soun et al., 2022) are standard. These resources demonstrate the potential of multimodality but are largely retrospective in nature, with text reflecting past conditions or summarizing known events rather than providing foresight for future forecasting.

Building on this foundation, recent benchmarks cover a broad spectrum of multimodal time series tasks. Early synthetic efforts (e.g., TS-Insights (Zhang et al., 2023), ChatTS (Xie et al., 2024), Context-aided Forecasting (Merrill et al., 2024)) generate captions or QA prompts that frequently

restate patterns already visible in the series. More "real-world" datasets such as TSQA (Kong et al., 2025a), MTBench (Chen et al., 2025), MoTime (Zhou et al., 2025), and Time-IMM (Chang et al., 2025) pair time series with textual context, images, or irregular sampling. Yet the text is often static, noisy, or weakly aligned with future outcomes, making it hard to evaluate whether models truly leverage auxiliary modalities for anticipatory reasoning rather than post hoc description.

Amid these efforts, Time-MMD (Liu et al., 2024a) and Context is Key (CiK) (Williams et al., 2025) have emerged as widely used benchmarks. Time-MMD's separation of textual facts vs. predictions is a step toward forecasting-oriented evaluation, but in practice the text can be incomplete or redundant, and causal links to future trajectories are often implicit, inviting spurious correlations. CiK emphasizes contextual grounding and event understanding, but its design primarily supports retrospective reasoning rather than explicit, scenario-based forecasting. These limitations motivate our benchmark: we provide expert-authored future scenarios that articulate plausible upcoming events, ensuring the textual modality carries genuine predictive value and enabling principled evaluation of multimodal models' ability to anticipate the future rather than merely describe the past.

## 2.2 MULTIMODAL FORECASTING APPROACHES

A broad range of multimodal forecasting studies have explored integrating textual and contextual signals with time series. Sociodojo (Cheng & Chin, 2024) and From News to Forecast (Wang et al., 2024) introduce agentic and reflective frameworks that process news, reports, and social media, while Xforecast (Aksu et al., 2024a) propose evaluation metrics for natural language explanations. Parallel efforts such as MetaTST (Dong et al., 2024), ContextFormer (Chattopadhyay et al., 2024), TextFusionHTS (Zhou et al., 2024), TaTS (Li et al., 2025b), LLMForecaster (Zhang et al., 2024), MLTA (Zhao et al., 2025), CHARM (Dutta et al., 2025), CAPTime (Yao et al., 2025), and SGCMA (Sun et al., 2025) enrich Transformer and hybrid architectures by incorporating metadata, textual descriptors, or probabilistic priors, demonstrating benefits for context-specific pattern learning and interpretability. Yet, these models remain constrained by the quality of textual inputs, which in existing benchmarks are often descriptive or redundant, rather than predictive of future outcomes.

More recent work has harnessed LLMs and generative paradigms. ChatTime (Wang et al., 2025b), DP-GPT4MTS (Liu et al., 2025), and TempoGPT (Zhang et al., 2025a) treat time series as a "language" or align temporal embeddings with text for reasoning-rich forecasting, while TimeXL (Jiang et al., 2025), Chronosteer (Wang et al., 2025a), and MCD-TSF (Su et al., 2025) employ LLM-in-the-loop refinement, instruction steering, or multimodal diffusion for probabilistic prediction. Time-VLM (Zhong et al., 2025) extends this line by leveraging visual signals. In addition, advanced prompting strategies (Ashok et al., 2025) can improve zero-shot context-aided forecasting, moving past simplistic prompting toward structured guidance that enables LLMs to better exploit auxiliary context. While these approaches showcase impressive reasoning and flexibility, their utility heavily depends on auxiliary text carrying genuine foresight; otherwise, their added complexity yields limited improvement over strong unimodal baselines.

Finally, some methods explicitly emphasize future-aware signals. The Multimodal Forecaster (Kim et al., 2024) jointly predicts time series and text, and the Dual Forecaster (Wu et al., 2025) integrates both historical descriptions and predictive future texts. Retrieval-augmented LLMs ground forecasts in historical corpora to mitigate hallucinations (Xiao et al., 2025). These works illustrate the promise of leveraging forward-looking or external knowledge, but their effectiveness is fundamentally constrained by benchmarks where text seldom provides actionable predictive content. Our benchmark addresses this gap by providing expert-authored future scenarios, ensuring that textual information is genuinely predictive and enabling principled evaluation of multimodal approaches across all these methodological families.

## 3 WHAT IF? TIME SERIES FORECASTING (WIT) BENCHMARK

### 3.1 PROBLEM SETUP

We consider a univariate time series $\{X_\tau\}_{\tau \geq 1}$ with $X_\tau \in \mathbb{R}$. Here, $\tau$ denotes the time index. At time $t$, the observed history is $x_{1:t} := (x_1, \ldots, x_t)$, and the forecasting task is to predict directional movement at horizon $h$, determined by the comparison between $x_t$ and $x_{t+h}$.

In addition to the raw time series, we assume access to textual context, which is divided into static context $S$ and dynamic context $D_t$. *Static context $S$* provides domain- or variable-level descriptions that remain fixed across time (e.g., definitions of approval rating measures in politics, descriptions of natural gas price indices in energy). *Dynamic context $D_t$* complements the historical observations by providing (i) historical context $H$, which explains past fluctuations not evident from $x_{1:t}$; (ii) future outlook $F_{\text{out}}$, which describes plausible scenarios for future trends; and (iii) counterfactual future $F_{\text{cf}}$, which specifies alternative hypothetical scenarios for counterfactual outcome. Here, $F_{\text{out}}$ and $F_{\text{cf}}$ both refer to the time horizon $t + 1$ to $t + h$. Concretely, future outlook provides forward-looking scenarios, often framed as *conditional statements* or *anticipated events*. And counterfactual future serves as a key test of whether models can adapt to signals beyond the observed history. Building on this setup, our benchmark assesses whether a model can perform future-conditioned forecasting using explicit outlooks $F_{\text{out}}$ or what-if scenarios $F_{\text{cf}}$, integrate multimodal evidence beyond only historical context, and generate conditional predictions of directional movement. Given the predictive distribution $q_{\theta,\phi}(y \mid x_{1:t}, S, H, F)$ over directional labels $y \in \{\text{rise}, \text{unchanged}, \text{fall}\}$, performance is measured by *directional accuracy (3-way)*, i.e., the proportion of correct directions.

## Types of Future Outlooks

**Conditional Statement** (Economy)

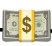

> "If unexpectedly soft economic data weaken near-term rate expectations, yields fall and risk appetite briefly improves, triggering safe-haven outflows and a short-term decline in the dollar broad index."

**Anticipated Event** (Politics)

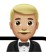

> "A national government will launch a broad job-creation initiative that will expand vocational training and increase employment opportunities."

Figure 2: Illustration of future outlook types, divided into conditional statements and anticipated events across domains.

### 3.2 TASK 1: TEXT-GUIDED SHORT TERM FORECASTING

Text-guided Short Term Forecasting focuses on forecasting over short horizons, where both the exact numerical value and the directional movement of the series are of practical importance. Given historical observations $x_{1:t}$ together with static and dynamic textual context $(S, H, F)$, the model is required to predict the immediate next step (or a few steps ahead) of the time series. Performance is evaluated by directional accuracy, and further complemented by the numerical precision of forecasts using MSE. This dual evaluation reflects that in short term forecasting, accurate values are often as meaningful as correctly identifying the trend direction.

### 3.3 TASK 2: TEXT-GUIDED LONG TERM FORECASTING

Text-guided Long Term Forecasting considers forecasting over longer horizons (e.g., several weeks ahead), where exact numerical values become increasingly uncertain. In such settings, the primary objective is not point-level accuracy but the ability to capture the overall directional trend relative to the last observed value. Accordingly, the task is evaluated by directional accuracy, which enhances reliable trend prediction under textual guidance rather than exact value matching.

### 3.4 TASK 3: TEXT-GUIDED COUNTERFACTUAL FORECASTING

Text-guided Counterfactual Forecasting evaluates whether models can faithfully follow counterfactual textual guidance. Counterfactual future is constructed by minimally altering the future context text, while keeping the historical time series and other contextual signals fixed. This minimal-change design, motivated by prior work on counterfactual reasoning (Wang et al., 2023; Youssef et al., 2024), ensures that any variation in prediction can be attributed solely to the modified guidance text. To avoid confounding long term dynamics, counterfactual evaluation follows the short term forecasting setup of Task 1.

Formally, the input remains $(x_{1:t}, S, H)$, but the future outlook $F_{\text{out}}$ is replaced with a counterfactual version $F_{\text{cf}}$. The evaluation criterion is inverted: the task assesses whether the predicted directional label $\hat{y}$ flips relative to the ground truth $y$, i.e., $\hat{y} \in \text{flip}(y)$ where $\text{flip}(y) = \{y' \in \{\text{rise}, \text{unchanged}, \text{fall}\} \mid y' \neq y\}$, $y \neq \text{unchanged}$. Cases with $y = \text{unchanged}$ are excluded. For example, $\text{flip}(\text{rise}) = \{\text{unchanged}, \text{fall}\}$ and $\text{flip}(\text{fall}) = \{\text{unchanged}, \text{rise}\}$.

Table 1: Comparison of multi-modal benchmarks for time series forecasting. ✓: available, ▲: partially available, ✗: not available. "variable-only" indicates cases where only variable descriptions are provided without richer contextual information.

| Datasets | Numerical | Static Context | Dynamic Context | | | Notes |
|---|---|---|---|---|---|---|
| | | Variable Description | Historical Analysis | Plausible Future | Counterfactual Scenario | |
| TS-Insights | ✓ | ▲ | ▲ | ✗ | ✗ | redundant with series |
| ChatTS | ✓ | ▲ | ▲ | ✗ | ✗ | overlaps with series |
| MoTime | ✓ | ✓ | ✗ | ✗ | ✗ | variable-only |
| MTBench | ✓ | ✓ | ▲ | ▲ | ✗ | noisy, inconsistent |
| TSQA | ✓ | ✓ | ▲ | ▲ | ✗ | raw or variable-only |
| Time-MMD | ✓ | ✓ | ▲ | ▲ | ✗ | incomplete, redundant |
| CiK | ✓ | ✓ | ✓ | ▲ | ✗ | raw + overly specific futures |
| WIT (ours) | ✓ | ✓ | ✓ | ✓ | ✓ | — |

## 4 DETAILS AND ANALYSIS OF THE WIT BENCHMARK

### 4.1 DOMAINS AND DATA SOURCES

WIT benchmark consists of four major domains: Politics, Society, Energy, and Economy, each combining structured time series with aligned textual data. In all cases, raw textual content was collected from reputable domestic and international news outlets as well as authoritative institutional reports, ensuring balanced and high-quality coverage across domains. The **Politics domain** combines crossnational approval ratings with diverse news narratives, offering a representative testbed for evaluating models under heterogeneous temporal and contextual conditions. In the **Society domain**, European housing price indices are paired with diverse news accounts, capturing how real estate dynamics intersect with broader social and economic contexts. In the **Energy domain**, Henry Hub natural gas prices are contextualized with agency reports and energy news, reflecting expert practices of integrating specialized analyses with contemporaneous media. In

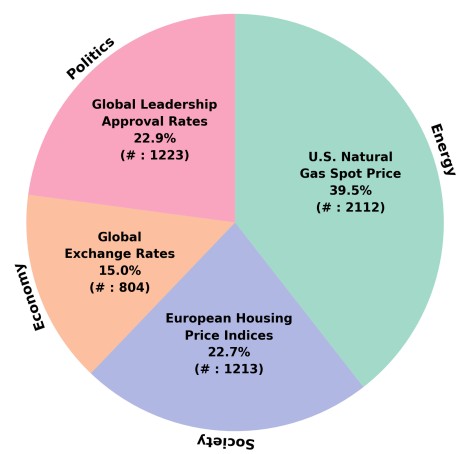

Figure 3: Overview of WIT benchmark domains and target variables, with domain-wise proportions and sample counts.

the **Economy domain**, the U.S. dollar index is paired with international media narratives on exchange rates and macroeconomic conditions, providing a comprehensive basis for assessing currency movements. By spanning politics, society, energy, and economy, the benchmark offers a diverse and complementary testbed that integrates quantitative signals with multifaceted textual context, enabling rigorous evaluation under the heterogeneous conditions encountered in practice. Further details on data sources are provided in Appendix A.3.

### 4.2 COMPARISON WITH EXISTING DATASETS

We compare existing multimodal time-series datasets and benchmarks with our WIT Benchmark in Table 1 across four dimensions. TS-Insights and ChatTS both contain textual information, yet this largely overlaps with raw numerical patterns. MoTime focuses mainly on merging modalities such as images, text, and time series, rather than providing dynamic contextual information. MTBench broadens coverage but introduces noisy and inconsistent text, lowering its reliability. Time-MMD and TSQA offer broader text, but much of it is incomplete, repetitive, or limited to variable descriptions, offering weak contextual signals. CiK uses raw information from periods unseen by LLMs, but updated models will learn such details—variable names, locations, and timestamps. Its historical information differs in nature: rather than offering contextual hints beyond the series, it provides deterministic summaries of past patterns, such as "*over the previous 90 days, the maximum sunlight occurred at 12:25:33 on average,*" effectively setting bounds rather than adding context. Its future texts often specify outcomes tied to exact dates, an unrealistic level of detail and overly specific.

Table 2: Results of controlled experiments evaluating the impact of information factors. Values represent accuracy. The highest average is indicated in **bold**, and the second-highest is underlined.

| Model | Short Term (Acc) | | | | Long Term (Acc) | | | |
|---|---|---|---|---|---|---|---|---|
| | History_TS | History_TS +History_CTX | History_TS +Future_OUT | History_TS +History_CTX +Future_OUT | History_TS | History_TS +History_CTX | History_TS +Future_OUT | History_TS +History_CTX +Future_OUT |
| Mistral-7B-Instruct | 0.441 | 0.498 | 0.445 | **0.535** | 0.469 | 0.498 | **0.615** | 0.600 |
| Qwen2.5-7B-Instruct | 0.501 | 0.502 | **0.772** | 0.768 | 0.502 | 0.497 | **0.768** | 0.734 |
| Gemma-3-27B-Instruct (4-bit) | 0.504 | 0.517 | **0.786** | 0.783 | 0.537 | 0.531 | **0.778** | 0.760 |
| Qwen3-32B (4-bit) | 0.512 | 0.519 | **0.786** | 0.778 | 0.522 | 0.524 | **0.783** | 0.748 |
| GPT-4o | 0.507 | 0.505 | **0.785** | 0.784 | 0.534 | 0.528 | **0.748** | 0.735 |

## 4.3 WHAT ARE THE KEY FACTORS IN TEXT-GUIDED TIME SERIES FORECASTING?

Previous benchmarks mainly focus on demonstrating the feasibility of text-guided TSF, but only few studies examine which factors are critical for dataset design. To address this gap, we conduct controlled experiments with representative LLMs, providing all factors in their original form and keeping prompt engineering to a minimum. We vary three components: time series, historical context, and future information. Multiple input configurations are constructed by selectively including or excluding each factor, and performance is compared across these settings.

Table 2 illustrates task accuracy for both short-term and long-term text-guided forecasting under different input conditions. The results show that incorporating future information consistently yields substantial improvements, both when used alongside time series data alone and when combined with time series and historical context. By contrast, historical context alone provides limited benefit; improvements are observed primarily when combined with future outlook. Despite careful curation, long historical text may dilute signal, and its utility can vary depending on how it is structured and presented to the model. These results identify future outlooks as the primary driver of text-guided TSF. Accordingly, WIT integrates future information with historical series and context. Appendix D provides experimental details and prompt templates.

## 4.4 DATA CONSTRUCTION PIPELINE

The WIT benchmark is built with a three-step pipeline. First, we form initial multimodal pairs. We collect timestamped text from daily and weekly reports and news headlines, then align each record to the corresponding time series by timestamp. These pairs serve as the foundation of the benchmark. Second, we refine the text with *a three-stage LLM process*: remove irrelevant or noisy content, align the narrative with actual series changes (e.g., computing point-wise deltas and using their signs to guide phrasing), and de-identify to avoid memorization leakage and encourage grounding in realistic causal drivers. Finally, we produce the WIT benchmark. For each sliding window with a historical interval followed by a future interval, we generate two contexts: a historical context written in the past tense that summarizes key events and observed impacts, and a future outlook written in scenario-based language that mirrors human forecasting practices, using conditional forms and modal verbs (e.g., "if," "may," "could") while avoiding explicit statements of impact.

For Task 3, we also generate counterfactual outlooks using a minimal-change strategy, following prior work that emphasizes small edits in synthetic counterfactuals (Youssef et al., 2024; Wang et al., 2023). We invert only a few key words (e.g., "increase" to "decrease") to preserve context while flipping directional implications coherently. This pipeline yields a dataset tightly aligned between time series and text, with reduced noise and minimal leakage. Further implementation details are in Appendix A.4.

## 4.5 MEMORIZATION MITIGATION AND DE-IDENTIFICATION

Since our corpus spans roughly the 2010s, modern LLMs (e.g., GPT-4o) may have been trained on overlapping facts such as specific companies, locations, dates, or named events. A naive alignment of raw text with time series could let models exploit memorized associations rather than reason over the provided context. We therefore keep the time-series signal intact and aligned, while de-identifying the aligned textual context so it preserves causal and mechanistic cues without direct lookup anchors. This preserves evaluative difficulty without drifting from the original dynamics. After integrating raw time series and text by exact timestamps, we build sliding windows to split

history and forecast horizons. During LLM-based text post-processing, we apply de-identification rules:

- **Temporal abstraction:** replace absolute dates and timestamps with relative references within the window (e.g., 'two days earlier,' 'in the prior week').

- **Entity masking:** replace specific companies, countries, regions, facilities, and event names with typed placeholders (e.g., [COMP_A], [REGION_B], [EVENT_C]).

- **Granularity control:** keep sector- or mechanism-level descriptors (supply shock, storage draw, policy guidance) that explain directionality, while removing uniquely identifying strings and URLs.

- **Consistency constraints:** ensure that the edited text remains temporally consistent with the windowed series (no future leakage, no contradictions to observed deltas).

Prior work (Williams et al., 2025) address memorization by using only the most recent information, generating derived series from raw data, or incorporating noise. These approaches can weaken alignment between text and series or push the task toward synthetic data. *Our approach keeps the real series and exact text–series alignment, while removing direct identifiers in the text.* This retains domain-faithful mechanisms that are useful for forecasting, without enabling trivial memorization.

Although concrete timestamps and names are removed, the text still conveys mechanism-level cues aligned to the series (e.g., supply disruptions, weather-driven demand, storage dynamics, policy stance). A model with genuine domain priors can still infer rise or fall logic from these cues, but cannot rely on database-like recall of a specific dated headline.

### 4.6 VALIDATING THE RELEVANCE OF THE CONTEXT

We validate that the textual context is relevant to forecasting and consistent with the aligned series. For each domain, human experts review sampled windows to verify that (i) the historical analyses and future scenarios are logically compatible with the observed series dynamics, and (ii) the forecast implied by the context would be reasonable given domain knowledge. For counterfactual instances, experts confirm that the text is constructed by minimal changes to the original scenario, and then assess whether the altered factor is plausibly causal for flipping the trend direction. This process ensures that both factual and counterfactual contexts are coherent, mechanism-grounded, and capable of justifying the ground-truth or counterfactually flipped outcomes.

## 5 EXPERIMENTS AND RESULTS

### 5.1 EXPERIMENTAL SETTING

We evaluate general-purpose LLMs, a task-aligned baseline (the instruction-tuned multimodal LLM for time series), state-of-the-art time series foundation models (TSFMs), and classical statistical methods on the WIT benchmark. Since WIT is designed purely for evaluation, it does not provide a training set. Therefore, we only consider models capable of producing forecasts without task-specific training.

**Scenario-guided Multimodal Forecasting** We evaluate both general-purpose LLMs and an instruction-tuned model for time series forecasting. The LLMs include Mistral-7B-Instruct-v0.3 (Jiang et al., 2023), Qwen2.5-7B-Instruct (Qwen et al., 2025), Gemma-3-27B-IT (Team et al., 2025), and Qwen3-32B (Yang et al., 2025), alongside the proprietary GPT-4o (OpenAI et al., 2024). All evaluations are conducted in a zero-shot setting with inputs consisting of time series data and associated text (data description, historical context, and future outlook). As a task-aligned baseline, we further include Time-MQA (Kong et al., 2025a), an instruction-tuned multimodal LLM that we denote as fine-tuned for time series (FTS). It is tuned on forecasting samples from the TSQA dataset, where data pairs numeric targets with trend tags, a format that closely matches WIT benchmark. Additional experimental details and prompt templates are provided in Appendix D.

**Unimodal (Time Series) Forecasting** As unimodal baselines, we evaluate both recent Transformer-based TSFMs and classical statistical methods. For TSFMs, we include

Table 3: Results of selected models on the WIT benchmark. Models are grouped into scenario-guided multimodal forecasting and unimodal time-series forecasting. The first two columns show Short Term task results (mean directional accuracy across all domains and MSE), followed by Long Term and Counterfactual tasks in mean directional accuracy. '–' indicates that counterfactual scenarios are not applicable to unimodal models, yielding the same outcome as the short-term task. An asterisk (*) in MSE denotes results reported exclusively for the Politics domain.

| Category | Task | Short Term | | Long Term | Counterfactual |
|---|---|---|---|---|---|
| | Model / Metric | MSE* | Acc | Acc | Acc |
| **Scenario-guided Multimodal Forecasting** | | | | | |
| LLMs | Mistral-7B-Instruct | 42.94 | 0.478 | 0.532 | 0.419 |
| | Qwen2.5-7B-Instruct | 29.07 | 0.890 | **0.693** | 0.896 |
| | Gemma-3-27B-Instruct (4-bit) | 20.82 | 0.864 | 0.675 | 0.867 |
| | Qwen3-32B (4-bit) | 22.75 | 0.869 | 0.685 | 0.909 |
| | GPT-4o | **13.49** | **0.919** | 0.645 | **0.969** |
| FTS | Time-MQA (Qwen2.5-7B) | 55.26 | 0.281 | 0.194 | 0.203 |
| **Unimodal (Time Series) Forecasting** | | | | | |
| TSFMs | Chronos-Bolt-Base | 17.99 | 0.529 | 0.526 | – |
| | Moirai-1.1-R-Large | 70.98 | 0.451 | 0.456 | – |
| | TimesFM-2.5-200M | 18.89 | 0.477 | 0.503 | – |
| Statistical | ARIMA | 382.7 | 0.385 | 0.419 | – |
| | ETS (State Space) | 31.71 | 0.539 | 0.551 | – |
| | Exponential Smoothing | 31.79 | 0.520 | 0.535 | – |

Chronos (Chronos-Bolt-Base) (Ansari et al., 2024), Moirai (Moirai-1.1-R-Large) (Woo et al., 2024), and TimesFM (TimesFM-2.5-200M) (Das et al., 2024), all tested in a zero-shot setting using only raw time series without domain-specific fine-tuning or textual inputs. For statistical methods, we consider ARIMA (Box & Jenkins, 1976), ETS (State Space) (Hyndman et al., 2008), and simple Exponential Smoothing (Brown, 2004), applied in a univariate setting with automatic configuration for trend and seasonality. Together, these unimodal baselines serve as a comparison point for WIT benchmark, highlighting the difference between models that use both text and time series and models that rely only on temporal patterns.

## 5.2 RESULTS ON WIT BENCHMARK

Table 3 summarizes the performance of all evaluated models across the three tasks of the WIT benchmark. LLMs that jointly leverage time series and textual descriptions substantially outperform unimodal TSFMs and classical statistical methods. This confirms the central motivation of WIT: leveraging scenario-guided textual context provides a clear advantage in both accuracy of short-term forecasts and alignment with counterfactual or outlook-based forecasting tasks. Notably, comparable short-term and counterfactual results show that models correctly leverage future text to differentiate opposing outcomes.

While Unimodal TSFMs and statistical methods capture historical regularities, they cannot utilize textual signals that encode anticipated events or hypothetical futures. These unimodal forecasting methods plateau in directional accuracy without external context. Unexpectedly, the poor transferability of Time-MQA emphasizes the importance of carefully designing instruction-tuning regimes for multimodal forecast-and-trend tasks. Together, these results highlight that WIT effectively distinguishes models capable of incorporating scenario-guided textual context from those limited to history series-only extrapolation.

## 5.3 Ablation Study

In constructing the historical context for WIT, we extract all significant events corresponding to the history time series without any restrictions. This process results in an average of 18.63 historical context per instance, which is a substantial amount. In the main experiments, we use these historical contexts directly as input without any additional processing. To investigate how the utilization of historical contexts affects model performance, we conduct a series of ablation experiments exploring different strategies for dynamically providing historical information.

We test four approaches. A manual recent filtering strategy (`recent4`) uses only the four most recent historical items as input, reflecting a human bias that recent events are most informative for predicting future trends. A random filtering strategy (`random4`) selects four items at random from the full set of historical contexts, serving as a contrast to manual selection and providing an unbiased, high-randomness baseline. Beyond these baselines, two LLM-guided strategies are also explored: in one variant (`llm_filter`), the model selects the four most important historical items, while in the other (`llm_summary`), the model generates a summary of the most critical historical information to use as input. These approaches evaluate the model's potential to process and leverage historical context effectively.

Table 4: Long-term forecast accuracy on the Politics domain of the WIT benchmark, comparing the performance of different historical context selection strategies. The best-performing results are underlined within each input configuration when historical context is included.

| Model | Method | Long Term (Acc) | | | |
|---|---|---|---|---|---|
| | | History_TS | History_TS +History_CTX | History_TS +Future_OUT | History_TS +History_CTX +Future_OUT |
| Qwen2.5-7B Instruct | default | 0.417 | 0.451 | 0.695 | 0.693 |
| | recent4 | - | 0.415 | - | 0.707 |
| | random4 | - | 0.420 | - | 0.695 |
| | llm_filter | - | 0.442 | - | 0.700 |
| | llm_summary | - | 0.407 | - | 0.695 |
| Qwen3-32B (4-bit) | default | 0.412 | 0.439 | 0.717 | 0.695 |
| | recent4 | - | 0.420 | - | 0.678 |
| | random4 | - | 0.451 | - | 0.688 |
| | llm_filter | - | 0.454 | - | 0.681 |
| | llm_summary | - | 0.434 | - | 0.698 |

As shown in Table 4, smaller model's performance tends to improve when using a manual filtering strategy (`recent4`) with the full input combination of history time series, historical context, and future outlook. As model size increased, performance is enhanced with LLM-guided strategies. However, no single strategy demonstrates universal superiority across all models. It indicates that in text-guided TSF, the optimal way of utilizing historical context can vary across models and depends critically on how the context is structured and presented. Refer to Appendix C.2 for more details.

## 6 Limitations and Future Work

While the WIT benchmark offers notable advances, several limitations remain. First, as historical context windows become longer, narratives describing upward and downward trends are often intermingled, introducing ambiguity that can hinder clear predictive guidance. Second, although de-identification and mitigation reduce leakage risks, some unique phrases may remain, and masking can reduce fine-grained information.

Looking forward, expanding both the scale and diversity of domains will be essential to strengthen generalization. Moreover, although the future outlook context provides clear directional guidance, its interplay with long historical contexts still requires closer examination. In particular, future work should explore how multimodal forecasting approaches can model the causal link between historical context and future outlook, making use of prospective signals embedded in past information as well as explicit future scenario cues provided by this benchmark. Beyond zero-shot evaluation, few-shot prompting also holds promise: by leveraging textual information about past dynamics as in-context examples, models may better capture how historical narratives inform plausible futures and counterfactual trajectories.

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

# Appendix

## Table of Contents

## A BENCHMARK DETAILS

### A.1 DATA STATISTICS

To provide a comprehensive overview of the constructed dataset, we summarize the number of instances across domains and task types in Table 5. The dataset covers four major domains: Politics, Society, Energy, and Economy, each of which is annotated under three distinct forecasting task settings: Short Term Forecasting, Long Term Forecasting, and Counterfactual Forecasting. This design ensures that the dataset not only reflects realistic domain diversity but also supports the evaluation of models under heterogeneous task conditions. Specifically, the Energy domain contains the largest number of instances (2,112 samples in total), reflecting the importance of high-frequency and long-horizon forecasting challenges in energy markets and environmental applications. In contrast, the Economy domain includes 804 samples, which are fewer in number but highlight complex interactions that arise in macroeconomic forecasting under limited contextual signals. Meanwhile, Politics (1,213 samples) and Society (1,223 samples) provide balanced coverage of socio-political contexts, particularly for scenarios where counterfactual reasoning (e.g., policy changes or social events) plays a crucial role. Overall, the dataset comprises 5,352 instances, with a relatively even distribution across domains. Importantly, the counterfactual setting accounts for nearly one-third of all samples, enabling systematic evaluation of models' robustness to alternative scenarios.

Table 5: Number of dataset instances across domains and tasks.

| Domain | Task | | | Total |
| --- | --- | --- | --- | --- |
| | Short Term | Long Term | Counterfactual | |
| Politics | 431 | 410 | 372 | 1213 |
| Society | 416 | 392 | 415 | 1223 |
| Energy | 705 | 702 | 705 | 2112 |
| Economy | 271 | 262 | 271 | 804 |

### A.2 DATA CHARACTERISTICS

Table 6: Number of time series data points provided as input and those to be predicted across horizons per domain.

| Domain | Window | Task | | |
| --- | --- | --- | --- | --- |
| | | Short Term | Long Term | Counterfactual |
| Politics | History | 8 | 8 | 8 |
| | Prediction | 1 | 4 | 1 |
| Society | History | 8 | 8 | 8 |
| | Prediction | 1 | 4 | 1 |
| Energy | History | 30 | 30 | 30 |
| | Prediction | 5 | 20 | 5 |
| Economy | History | 30 | 90 | 30 |
| | Prediction | 20 | 30 | 20 |

Table 6 presents the number of input and predicted time series points across domains. The configuration reflects both domain characteristics and typical prediction durations. In Politics and Society domain, where data are recorded at coarser intervals (weekly, monthly, or quarterly), using many historical points would correspond to an excessively long temporal span. Consequently, shorter input windows are employed to provide a manageable history length. Conversely, Energy and Economy domains primarily consist of daily data, where the same number of points represents a shorter temporal span, allowing longer input windows and extended prediction horizons to capture higher-frequency dynamics. This design ensures that the forecasting setup is aligned with both the temporal resolution and practical predictive requirements of each domain.

Table 7: Average number of sentences and tokens per component.

| Component | Avg # of Sentences | Avg # of Tokens |
|---|---|---|
| Historical Context | 18.63 | 479.25 |
| Future Outlook | - | 52.30 |
| Data Description | - | 68.31 |

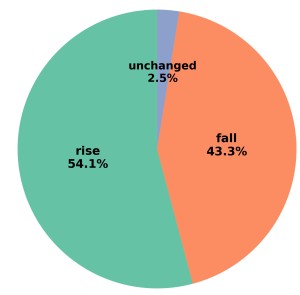

Figure 4: Class distribution of WIT benchmark.

Table 7 reports the average number of sentences and tokens for each data component in the WIT benchmark. Historical context contains the largest volume of text per instance, with an average of 18.63 sentences and 479.25 tokens, whereas future outlook and data description are comparatively shorter. Figure 4 illustrates the class distribution for the Long Term Forcasting task. Due to the high volatility characteristic of the time series data, instances labeled as "unchanged" are scarce, while "rise" and "fall" labels appear in roughly balanced proportions, reflecting a reasonable class distribution of WIT benchmark.

### A.3 DETAILS OF DATA SOURCES AND VARIABLES BY DOMAIN

#### A.3.1 POLITICS DOMAIN DATASET SOURCES

In the Politics domain, the time series data capture approval ratings of national leaders across multiple countries, reflecting diverse political systems and regional contexts. These series are complemented by rich textual narratives from a wide range of international and domestic news organizations, enabling the benchmark to cover not only different leaders and administrations but also varied media perspectives and reporting traditions. This diversity ensures that the political domain provides a broad and representative basis for evaluating models under heterogeneous temporal and contextual conditions.

**Time Series Data** The time series data consist of approval ratings of national leaders across multiple countries, collected at varying intervals. The raw data can be accessed from the Statista website[1].

**Text Data** The raw textual data are collected from major domestic and international news outlets. To mitigate potential bias, all personal names, geographic references, and other identifiable information were carefully anonymized during preprocessing.

**Source 1 : United States**

TIME SERIES DATA

- Gallup, Do you approve or disapprove of the way Barack Obama is handling his job as president?
  https://www.statista.com/statistics/205284/obama-job-approval-rate-by-the-american-public/
- Gallup, Donald Trump presidential approval rating in the United States from 2017 to 2021, and 2025
  https://www.statista.com/statistics/666113/approval-rate-of-donald-trump-for-the-presidential-job/
- YouGov, Monthly presidential job approval rating of Joe Biden in the United States from 2021 to 2025

---

[1]https://www.statista.com/

```
https://www.statista.com/statistics/1222960/
approval-rate-monthly-joe-biden-president/
```

TEXT DATA

- Source : The New York Times[2], The Washington Post[3], Reuters[4], NPR[5], AP[6]
- Type : News articles selected based on domain relevance and keyword filtering.
- Coverage : From January 2009 to January 2025

**Source 2 : Canada**

TIME SERIES DATA

- Angus Reid Institute, Domestic approval and disapproval rating of Canadian Prime Minister Justin Trudeau from September 2014 to February 2025
  ```
  https://www.statista.com/statistics/1600839/
  justin-trudeau-canada-approval-rating/
  ```

TEXT DATA

- Source : CBC[7], The Globe and Mail[8], National Post[9], The Guardian[10], AP
- Type : News articles selected based on domain relevance and keyword filtering.
- Coverage : From September 2014 to February 2025

**Source 3 : Republic of Korea**

TIME SERIES DATA

- Gallup Korea, Approval rating of South Korea's President Yoon Suk Yeol from April 2022 to December 2024
  ```
  https://www.statista.com/statistics/1311511/
  south-korea-approval-rating-of-president-yoon-suk-yeol/
  ```

TEXT DATA

- Source : The Chosun Daily[11], The Joongang[12], Hankyoreh[13], The Guardian, Reuter
- Type : News articles selected based on domain relevance and keyword filtering.
- Coverage : From April 2022 to December 2024

**Source 4 : Japan**

---

[2] https://www.nytimes.com/
[3] https://www.washingtonpost.com/
[4] https://www.reuters.com/
[5] https://www.npr.org/
[6] https://apnews.com/
[7] https://www.cbc.ca/
[8] https://www.theglobeandmail.com/
[9] https://nationalpost.com/
[10] https://www.theguardian.com/international
[11] https://www.chosun.com/
[12] https://www.joongang.co.kr/
[13] https://www.hani.co.kr/

TIME SERIES DATA

- NHK, Monthly approval ratings for the cabinet in Japan from January 2019 to June 2025
  `https://www.statista.com/statistics/1263388/`
  `japan-monthly-cabinet-approval-rating/`

TEXT DATA

- Source : NHK[14], The Asahi Shimbun[15], The Mainichi[16], The Guardian, Reuter
- Type : News articles selected based on domain relevance and keyword filtering.
- Coverage : From January 2019 to June 2025

**Source 5 : France**

TIME SERIES DATA

- IFOP, Do you approve or disapprove of Emmanuel Macron's actions as President of France?
  `https://www.statista.com/statistics/941208/`
  `macron-approval-ratings/`

TEXT DATA

- Source : Le Figaro[17], Le Monde[18], France 24[19], The Guardian, Reuter
- Type : News articles selected based on domain relevance and keyword filtering.
- Coverage : From May 2017 to February 2024

### A.3.2 SOCIETY DOMAIN DATASET SOURCES

The Society domain focuses on real estate markets, represented by quarterly house price indices across a wide range of European countries. These quantitative signals are complemented by textual narratives from diverse national news outlets that cover housing, financial, and broader social conditions. By integrating structured indicators with varied media perspectives across different regions, this domain provides a rich setting for examining how societal and economic developments are reflected jointly in time series and text.

**Time Series Data**    The time series data comprise house price indices for multiple European countries, collected quarterly. The raw dataset is accessible through the Statista website.

- Bank for International Settlements, Quarterly house price index (inflation-adjusted) in select countries in Europe from 3rd quarter 2010 to 4th quarter 2024
  `https://www.statista.com/statistics/722946/`
  `house-price-index-in-real-terms-in-eu-28/`

**Text Data**    The text data originate from multiple domestic news outlets and were collected based on domain relevance and keyword filtering. The data cover events between July 2017 and December 2024, aligning with the time span of the time series data. The country-specific sources are listed below.

---

[14]`https://www.nhk.or.jp/`
[15]`https://www.asahi.com/`
[16]`https://mainichi.jp/`
[17]`https://www.lefigaro.fr/`
[18]`https://www.lemonde.fr/`
[19]`https://www.france24.com/`

- Ireland : RTÉ[20], The Irish Times[21], The Irish Independent[22]
- Spain : El País[23], ABC[24], El Mundo[25]
- Switzerland : SRF[26], Tages-Anzeiger[27], NZZ[28]
- Estonia : Postimees[29], Eesti Paevaleht[30], Maaleht[31]
- Hungary : Magyar Nemzet[32], Nepszava[33], 24.hu[34]
- Germany : Der Spiegel[35], Die Zeit[36], Frankfurter Allgemeine Zeitung[37]
- Belgium : Le Soir[38], De Standaard[39], Het Laatste Nieuws[40]

### A.3.3 ENERGY DOMAIN DATASET SOURCES

In the Energy domain, the dataset centers on natural gas markets, with the Henry Hub spot price serving as the primary time series indicator. To complement these quantitative signals, we draw on multiple forms of authoritative textual context, including daily and weekly reports from trusted energy agencies as well as broad coverage of energy-related news filtered for relevance to natural gas. This design mirrors the way domain experts would gather and synthesize information from specialized institutional analyses and contemporaneous media reporting, thereby assembling the diverse materials necessary for informed forecasting and decision-making in complex energy markets.

**Time Series Data** The time series data consist of Henry Hub Natural Gas Spot Price (NG.RNGWHHD.D) obtained from the U.S. Energy Information Administration (EIA) through its Open Data API[41].

**Text data** The raw textual data are collected from official reports from the U.S. Energy Information Administration (EIA) as well as major domestic and international energy news headlines from Global Database of Events, Language, and Tone (GDELT) project. To mitigate potential bias, specific dates, company names, facility locations, and other identifiable information were carefully anonymized during preprocessing.

- (Daily reports) U.S. EIA, Today in Energy tagged with Natural Gas from February 9, 2011 to September 11, 2025
  `https://www.eia.gov/todayinenergy/index.php?tg=natural%20gas`
- (Weekly reports) U.S. EIA, Natural Gas Weekly Update from January 6, 2011 to September 4, 2025
  `https://www.eia.gov/naturalgas/weekly/`

---

[20]`https://www.rte.ie/`
[21]`https://www.irishtimes.com/`
[22]`https://www.independent.ie/`
[23]`https://elpais.com/?ed=es`
[24]`https://www.abc.es/`
[25]`https://www.elmundo.es/`
[26]`https://www.srf.ch/`
[27]`https://www.tagesanzeiger.ch/`
[28]`https://www.nzz.ch/`
[29]`https://www.postimees.ee/`
[30]`https://epl.delfi.ee/`
[31]`https://maaleht.delfi.ee/`
[32]`https://magyarnemzet.hu/`
[33]`https://nepszava.hu/`
[34]`https://24.hu/`
[35]`https://www.spiegel.de/`
[36]`https://www.zeit.de/index`
[37]`https://www.faz.net/aktuell/`
[38]`https://www.lesoir.be/`
[39]`https://www.standaard.be/`
[40]`https://www.hln.be/`
[41]`https://www.eia.gov/opendata/`

- (News headlines) GDELT 1.0 Global Knowledge Graph (GKG) from April 1, 2013 to September 18, 2025 [Keywords: natural gas or Henry Hub]
  `http://data.gdeltproject.org/gkg/index.html`

### A.3.4  ECONOMY DOMAIN DATASET SOURCES

The dataset in the Economy domain centers on the Nominal Broad U.S. Dollar Index, a key indicator of global financial conditions. To contextualize fluctuations in this target variable, we incorporate a wide range of textual data that capture discussions of exchange rates, dollar strength, and related macroeconomic developments across international media sources. By aggregating diverse reports and articles filtered around the dollar index, the dataset reflects the type of comprehensive information landscape that human experts would consult when forming judgments about currency movements. This integration ensures that the economic domain provides not only structured market signals but also the broader contextual narratives needed for realistic forecasting and decision-making.

**Time Series Data**  The time series data consist of the Nominal Broad U.S. Dollar Index (DTWEXBGS) obtained from the Federal Reserve Bank of St. Louis (FRED) through the website [42].

**Text data**  The raw textual data are collected from from major domestic and international news outlets. To mitigate potential bias, all identifiable information were carefully anonymized during preprocessing.

- (News) GDELT 1.0 Global Knowledge Graph (GKG) from April 1, 2013 to March 31, 2024 [Keywords: dollar index, USD index, DXY, exchange rate]
  `http://data.gdeltproject.org/gkg/index.html`

### A.4  DATA CONSTRUCTION PIPELINE

### A.4.1  DETAILS ON PIPELINE

GPT APIs were employed for both data refinement and the construction of the WIT benchmark. GPT-4o-mini was used during the refinement stage, whereas GPT-5-mini handled the data generation stage. This separation allowed the pipeline to leverage the strengths of each model, ensuring high-quality and consistent textual data. After generation, all data were thoroughly double-checked by domain experts. For counterfactual future instances, a rule-based validation was first applied where possible, followed by expert review, further ensuring the reliability and quality of the benchmark.

### A.4.2  PROMPT TEMPLATES

The following prompt templates were used to generate the main components of the WIT benchmark: historical context, future outlook, and counterfactual future. In these templates, `domain_adj` specifies the domain (e.g., political, societal), granularity corresponds to the time interval at which the series was collected (e.g., week, month, quarter), `target_variable` represents the specific metric being predicted (e.g., approval rate, natural gas spot price), and `events` contains curated event summaries corresponding to the historical time series. These structured templates ensured consistent and domain-relevant generation of textual context across the benchmark.

---

[42]`https://fred.stlouisfed.org/series/DTWEXBGS`

Table 8: Prompt template used for generating historical context in WIT benchmark.

```
    prompt = f"""
You are given historical {DOMAIN_ADJ} event summaries with {GRANULARITY}ly
{TARGET_VARIABLE} changes.

Instructions:
- Carefully review each event summary.
- If a summary contains multiple important issues, split them and summarize each one
separately.
- Select only the issues most likely to have affected the {TARGET_VARIABLE}.
- Summarize each issue and its impact in 1 concise sentence in the past tense.
- Match the tone provided for each entry.
- Return each sentence as a separate bullet.
- Do not start sentences with temporal phrases.
- Do not mention {TARGET_VARIABLE}, numbers, or speculation.
- Do not ask for clarification or additional information.

Historical events with tone:
{EVENTS}
"""
```

Table 9: Prompt template used for generating future outlook in WIT benchmark.

```
    prompt = f"""
You are given summaries of future {DOMAIN_ADJ} events with {GRANULARITY}ly
{TARGET_VARIABLE} changes.

Instructions:
- Select the single most significant sub-event among the summaries.
- Summarize it in one short future-tense sentence.
- Match the tone hint provided for the chosen sub-event.
- Include only the core point.
- Do not mention speculation or interpretation.
- Do not mention {TARGET_VARIABLE}.
- Do not ask for clarification or additional information.

Future events with tone:
{EVENTS}
"""
```

Table 10: Prompt template used for generating counterfactual future in WIT benchmark.

```
    prompt = f"""
You are given a {DOMAIN_ADJ} event summary: "{TEXT}"
Create a counterfactual version of this event by reversing the main event.

Instructions:
- Keep the description plausible and in the same style.
- Include only the main reversal of the event; do not add extra details.
- Do not ask for clarification or additional information.
- Return only the counterfactual text.
"""
```

Table 11: Code example used for validating counterfactual future in WIT benchmark.

```python
def validate_counterfactual_logic(original: str, counterfactual: str) -> bool:
    """
    Validate if the counterfactual text is logically consistent with the original.
    Returns True if valid, False otherwise.
    """

    # 1. Check for contradictory conditions
    contradictions = [
        ("falling yields", "flows into"),
        ("rising yields", "flows from"),
        ("loose conditions", "tightening"),
        ("tight conditions", "easing"),
        ("economic weakness", "dollar strength"),
        ("economic strength", "dollar weakness")
    ]

    for condition, outcome in contradictions:
        if condition.lower() in counterfactual.lower() and outcome.lower() in
        counterfactual.lower():
            return False

    # 2. Check that key terms are properly changed (should not remain the same)
    unchanged_pairs = [
        ("safe-haven", "safe-haven"),          # should be changed
        ("carry flows from", "carry flows from")  # should be inverted
    ]
    for original_term, cf_term in unchanged_pairs:
        if original_term.lower() in original.lower() and cf_term.lower() in
        counterfactual.lower():
            return False

    # 3. Check policy timing consistency: only direction changes, timing stays
    if "later easing" in original.lower() and "earlier tightening" in
    counterfactual.lower():
        return False
    if "earlier easing" in original.lower() and "later tightening" in
    counterfactual.lower():
        return False

    # 4. Economic logic check: safe-haven should not remain unchanged
    if "safe-haven" in original.lower() and "safe-haven" in counterfactual.lower():
        return False

    return True
```

# B  DATA SAMPLE

## B.1  POLITICS DOMAIN

Table 12: Illustrative data sample for the Text-Guided Short Term Forecasting task.

```
{
  "domain": "Politics",
  "task": "Text Guided Short Term Forecasting",
  "description": {
    "task_description": "The task is to predict the target variable for the next
    step.",
    "data_description": "The following data is from the political domain and
    contains presidential approval ratings. Approval ratings range from 0 to 100
    and reflect public responses to various political and social events, policies,
    and issues. For reference, the average change between consecutive time points
    in approval ratings is 3.6842. "
  },
  "data": {
    "history_timeseries": [66, 60, 56, 46, 44, 44, 50, 53],
    "historical_context_text": [
      "- A newly elected president was inaugurated and swiftly formed a centrist
      administration that drew personnel from across the political spectrum.",
      (...)
      "- The administration advanced pro-business reforms and tax-relief policies
      while maintaining relative political stability and cooperative relations with
      regional partners."
    ],
```

```
      "future_outlook_text": "An organization will implement large-scale layoffs and
      disclose a substantial budget shortfall.",
      "prediction_horizon": 1
    },
    "answer": {
      "future_timeseries": [48.0],
      "trend": "fall"
    }
  }
```

Table 13: Illustrative data sample for the Text-Guided Long Term Forecasting task.

```
  {
    "domain": "Politics",
    "task": "Text Guided Long Term Forecasting",
    "description": {
      "task_description": "The task is to classify the target variable trend 4 steps
      ahead compared to the last data point as one of: rise, unchanged, or fall.",
      "data_description": "The following data is from the political domain and
      contains prime minister approval ratings. Approval ratings range from 0 to 100
      and reflect public responses to various political and social events, policies,
      and issues. For reference, the average change between consecutive time points
      in approval ratings is 3.9194. "
    },
    "data": {
      "history_timeseries": [35, 33, 32, 31, 33, 35, 36, 43],
      "historical_context_text": [
        "- Falling energy prices and stalled pipeline projects deepened regional
        economic hardship and provoked public frustration.",
        (...)
        "- Ongoing job growth and low unemployment demonstrated steady economic
        performance and reinforced public confidence in federal leadership."
      ],
      "future_outlook_text": [
        "A prominent institution will face intensified scrutiny as systemic failures
        deepen.",
        "A major reform program will deliver noticeable improvements in services and
        economic performance."
      ],
      "prediction_horizon": 4,
      "options": ["rise", "unchanged", "fall"]
    },
    "answer": {
      "trend": "rise",
      "full_future_timeseries": [33, 54, 55, 50]
    }
  }
```

Table 14: Illustrative data sample for the Text-Guided Counterfactual Forecasting task.

```
  {
    "domain": "Politics",
    "task": "Text Guided Counterfactual Forecasting",
    "description": {
      "task_description": "The task is to classify the target variable trend 1 step
      ahead compared to the last data point as one of: rise, unchanged, or fall.",
      "data_description": "The following data is from the political domain and
      contains cabinet approval ratings. Approval ratings range from 0 to 100 and
      reflect public responses to various political and social events, policies, and
      issues. For reference, the average change between consecutive time points in
      approval ratings is 3.9221. "
    },
    "data": {
      "history_timeseries": [42, 47, 48, 48, 45, 49, 48, 47],
      "historical_context_text": [
        "- Allegations of cronyism and a linked documentfalsification scandal
        undermined public trust in the government's competence and integrity.",
        (...)
```

```
        "- Weak economic data and criticism over the government's handling of
        recovery from a major storm compounded dissatisfaction with its performance."
      ],
      "future_outlook_text": "Economic conditions will improve, triggering lower
      unemployment and stronger household finances.",
      "prediction_horizon": 1
    },
    "answer": {
      "trend": ["rise", "unchanged"]
    }
  }
```

## B.2 SOCIETY DOMAIN

Table 15: Illustrative data sample for the Text-Guided Short Term Forecasting task.

```
  {
    "domain": "Society",
    "task": "Text Guided Short Term Forecasting",
    "description": {
      "task_description": "The task is to predict the target variable for the next
      step.",
      "data_description": "The following data is from the societal domain and
      contains house price indices. House price indices are standardized to 100 in
      the base year, and subsequent values represent relative changes. They capture
      the impact of societal events, including policies, economic developments, and
      other relevant factors. For reference, the average change between consecutive
      time points in the house price index is 2.3055. "
    },
    "data": {
      "history_timeseries": [106.19, 101.79, 98.28, 93.83, 88.9, 83.16, 78.12, 73.4],
      "historical_context_text": [
        "- A deep recession, rising unemployment and large banking recapitalisations
        alongside fiscal consolidation squeezed incomes and investor confidence,
        which depressed housing demand.",
        (...)
        "- Severe credit constraints from bank restructuring, high unemployment and
        emigration, and rising arrears and repossessions sharply curtailed demand."
      ],
      "future_outlook_text": "Rising unemployment will sharply reduce consumer
      spending and push many households into financial distress.",
      "prediction_horizon": 1
    },
    "answer": {
      "future_timeseries": [69.92],
      "trend": "fall"
    }
  }
```

Table 16: Illustrative data sample for the Text-Guided Long Term Forecasting task.

```
  {
    "domain": "Society",
    "task": "Text Guided Long Term Forecasting",
    "description": {
      "task_description": "The task is to classify the target variable trend 4 steps
      ahead compared to the last data point as one of: rise, unchanged, or fall.",
      "data_description": "The following data is from the societal domain and
      contains house price indices. House price indices are standardized to 100 in
      the base year, and subsequent values represent relative changes. They capture
      the impact of societal events, including policies, economic developments, and
      other relevant factors. For reference, the average change between consecutive
      time points in the house price index is 0.9787. "
    },
    "data": {
      "history_timeseries": [103.02, 103.73, 104.14, 107.55, 106.8, 106.58, 107.99,
      110.08],
      "historical_context_text": [
```

```
        "- A government collapse over a contentious migration agreement and ensuing
        political uncertainty dented consumer confidence and market sentiment.",
        (...)
        "- Ultralow interest rates and favorable mortgage conditions, together with
        limited housing supply, supported robust buyer activity."
      ],
      "future_outlook_text": [
        "Major policy measures will boost consumer and investor confidence and spur
        demand.",
        "Rising investment and business expansion will accelerate local economic
        activity and job creation."
      ],
      "prediction_horizon": 4,
      "options": ["rise", "unchanged", "fall"]
    },
    "answer": {
      "trend": "rise",
      "full_future_timeseries": [112.44, 112.61, 113.92, 115.37]
    }
  },
```

Table 17: Illustrative data sample for the Text-Guided Counterfactual Forecasting task.

```
{
  "domain": "Society",
  "task": "Text Guided Counterfactual Forecasting",
  "description": {
    "task_description": "The task is to classify the target variable trend 1 step
    ahead compared to the last data point as one of: rise, unchanged, or fall.",
    "data_description": "The following data is from the societal domain and
    contains house price indices. House price indices are standardized to 100 in
    the base year, and subsequent values represent relative changes. They capture
    the impact of societal events, including policies, economic developments, and
    other relevant factors. For reference, the average change between consecutive
    time points in the house price index is 2.9308. "
  },
  "data": {
    "history_timeseries": [187.31, 184.47, 186.25, 185.42, 189.93, 190.76, 193.42,
    191.2],
    "historical_context_text": [
      "- Soaring inflation and sharply higher energy costs eroded household real
      incomes and reduced purchasing power.",
      (...)
      "- Domestic fiscal tightening and stricter mortgage or regulatory measures,
      together with a drop in foreign buyer interest, further depressed demand."
    ],
    "future_outlook_text": "Unemployment will fall and credit conditions will
    loosen, alleviating economic strain.",
    "prediction_horizon": 1
  },
  "answer": {
    "trend": ["rise", "unchanged"]
  }
}
```

## B.3 ENERGY DOMAIN

Table 18: Illustrative data sample for the Text-Guided Short Term Forecasting task.

```
{
  "domain": "Energy",
  "task": "Text Guided Short Term Forecasting",
  "description": {
    "task_description": "The task is to predict the target variable for the next 5
    steps.",
```

```
    "data_description": "The following data is from the Energy domain and contains
    Henry Hub natural gas spot prices observed in winter. Seasonal variation is
    important, as demand patterns in winter and summer significantly affect natural
    gas consumption and market dynamics. In addition, production levels and storage
    inventories are critical factors that influence overall supply conditions and
    market behavior."
},
"data": {
    "history_timeseries": [4.52, 4.49, 4.42, 4.49, 4.42, 4.55, 4.48, 4.38, 4.52,
    4.48, 4.57, 4.72, 4.72, 4.46, 4.40, 4.41, 4.27, 4.42, 4.42, 4.55, 4.69, 4.48,
    4.32, 4.24, 4.22, 4.11, 3.96, 3.89, 3.92, 3.93],
    "historical_context_text": [
        "Natural gas spot prices increased across all domestic pricing points,
        influenced by rising demand for heating amid colder-than-normal
        temperatures.",
        (...)
        "Overall, the interplay of weather conditions, supply constraints, and
        regulatory changes presents a complex landscape for short-term forecasting in
        the natural gas market."
    ],
    "future_outlook_text": [
        "If temperatures remain above average, demand for heating will likely
        decrease, leading to further declines in market conditions."
    ],
    "prediction_horizon": 5,
    "options": ["rise", "unchanged", "fall"]
},
"answer": {
    "future_timeseries": [3.9, 3.84, 3.89, 3.83, 3.83],
    "trend": "fall"
}
}
```

Table 19: Illustrative data sample for the Text-Guided Long Term Forecasting task.

```
{
    "domain": "Energy",
    "task": "Text Guided Long Term Forecasting",
    "description": {
        "task_description": "The task is to classify the target variable trend 20 steps
        ahead compared to the last data point as one of: rise, unchanged, or fall.",
        "data_description": "The following data is from the Energy domain and contains
        Henry Hub natural gas spot prices in winter. Seasonal variation is important,
        as demand patterns in winter and summer significantly affect natural gas
        consumption and market dynamics. In addition, production levels and storage
        inventories are critical factors that influence overall supply conditions and
        market behavior."
    },
    "data": {
        "history_timeseries": [2.43, 2.42, 2.31, 2.17, 2.18, 2.28, 2.28, 2.28, 2.34,
        2.30, 2.26, 2.21, 2.27, 2.17, 2.11, 2.11, 2.09, 1.75, 2.06, 2.09, 2.05, 2.06,
        2.10, 2.17, 2.09, 2.05, 2.05, 2.03, 2.15, 2.01],
        "historical_context_text": [
            "The U.S. Energy Information Administration updated geologic maps of a key
            formation, enhancing understanding of regional production potential.",
            (...)
            "The anticipated growth in renewable energy capacity may impact natural gas
            demand dynamics in the coming years."
        ],
        "future_outlook_text": [
            "Assuming warmer-than-usual temperatures persist, residential and commercial
            natural gas consumption may decline, leading to reduced demand for heating.",
            "With ongoing maintenance on key pipelines, natural gas exports to
            neighboring markets could face interruptions, further contributing to a
            decrease in overall market activity."
        ],
        "prediction_horizon": 20,
        "options": ["rise", "unchanged", "fall"]
    },
    "answer": {
        "trend": "fall",
        "full_future_timeseries": [2.06, 2.07, 1.98, 1.89, 1.95, 1.91, 2.03, 1.96,
```

```
    1.93, 1.94, 1.91, 1.90, 1.89, 1.89, 1.86, 1.93, 1.85, 1.85, 1.91, 1.95]
  }
},
```

Table 20: Illustrative data sample for the Text-Guided Counterfactual Forecasting task.

```
{
  "domain": "Energy",
  "task": "Text Guided Counterfactual Forecasting",
  "description": {
    "task_description": "The task is to classify the target variable trend 5 steps
    ahead compared to the last data point as one of: rise, unchanged, or fall.",
    "data_description": "The following data is from the Energy domain and contains
    Henry Hub natural gas spot prices in summer. Seasonal variation is important,
    as demand patterns in winter and summer significantly affect natural gas
    consumption and market dynamics. In addition, production levels and storage
    inventories are critical factors that influence overall supply conditions and
    market behavior."
  },
  "data": {
    "history_timeseries": [3.10, 3.24, 3.24, 3.20, 3.08, 3.11, 3.22, 3.21, 3.31,
    3.42, 3.52, 3.50, 3.50, 3.16, 3.08, 3.13, 3.10, 3.12, 3.08, 2.98, 2.99, 3.00,
    2.89, 2.98, 3.02, 3.05, 3.03, 3.05, 2.93, 2.95],
    "historical_context_text": [
      "Increased energy consumption in the region indicates a growing demand for
      natural gas, driven by higher temperatures and cooling degree days.",
      (...)
      "The evolving energy trade landscape, including tariffs and international
      agreements, is reshaping the dynamics of U.S. energy exports."
    ],
    "future_outlook_text": [
      "If there is a sudden rise in power demand due to unseasonably warm weather,
      supply could lag behind consumption, leading to tighter market conditions."
    ],
    "prediction_horizon": 5,
    "options": ["rise", "unchanged", "fall"]
  },
  "answer": {
    "trend": ["rise", "unchanged"]
  }
},
```

## B.4 ECONOMY DOMAIN

Table 21: Illustrative data sample for the Text-Guided Short Term Forecasting task.

```
{
  "domain": "Economy",
  "task": "Text Guided Short Term Forecasting",
  "description": {
    "task_description": "The task is to predict the target variable for the next 5
    steps.",
    "data_description": "The following data is from the economy domain and contains
    the U.S. dollar broad index (DTWEXBGS). Daily variation is important, as
    short-term shocks often arise from economic releases, monetary policy
    expectations, and geopolitical events, while structural drivers such as trade
    flows and capital markets shape baseline conditions."
  },
  "data": {
    "history_timeseries": [97.1711, 97.4144, 97.3943, 97.3532, 97.2869, 97.3067,
    97.2874, 97.2488, 97.4590, 97.6047, 97.4772, 97.5423, 97.2173, 97.0145,
    97.4560, 98.1163, 98.5336, 98.5576, 98.8086, 99.1325, 99.0236, 98.9305,
    98.8897, 99.1045, 99.0917, 99.2188, 99.0185, 99.2424, 99.2554, 99.2894],
```

```
      "historical_context_text": [
        "Government bond yields and interestrate expectations alternated between
        firming and easing, affecting currency demand.",
        (...)
        "Plunging crudeoil and commodity prices pressured commoditylinked currencies
        and risksensitive sectors."
      ],
      "future_outlook_text": [
        "If policy communication turns unexpectedly hawkish and lifts near-term rate
        expectations, yields rise and funding tightens, prompting carry and
        funding-driven flows into the dollar that boost the broad index."
      ],
      "prediction_horizon": 5,
      "options": ["rise", "unchanged", "fall"]
    },
    "answer": {
      "future_timeseries": [99.4227, 99.2837, 99.1893, 100.0719, 99.8964],
      "trend": "rise"
    }
  }
```

Table 22: Illustrative data sample for the Text-Guided Long Term Forecasting task.

```
{
  "domain": "Economy",
  "task": "Text Guided Long Term Forecasting",
  "description": {
    "task_description": "The task is to classify the target variable trend 30 steps
    ahead compared to the last data point as one of: rise, unchanged, or fall.",
    "data_description": "The following data is from the economy domain and contains
    the U.S. dollar broad index (DTWEXBGS). Over longer horizons, persistent
    factors such as global monetary policy divergence, capital flows, and
    macroeconomic fundamentals dominate, while transient shocks average out."
  },
  "data": {
    "history_timeseries": [117.5552, 117.7423, 117.6854, 117.2820, 116.8257,
    116.8122, 116.6438, 116.2631, 116.1359, 116.0453, 115.9868, 116.0601,
    116.1291, 116.0788, 116.0236, 115.9811, 116.1649, 115.9137, 115.7324,
    115.8637, 116.1149, 116.1012, 116.1179, 116.3440, 116.5843, 116.8404,
    116.8031, 116.4409, 116.3634, 116.5402, 116.8293, 116.7447, 116.9148,
    117.0109, 117.0529, 117.1292, 117.1218, 116.9664, 116.8916, 116.6938,
    116.3857, 116.5394, 116.3296, 116.2557, 116.1492, 115.8755, 115.6976,
    115.5559, 115.5561, 115.6627, 115.5997, 115.7604, 115.8066, 115.6347,
    115.2207, 114.9639, 114.6697, 114.9746, 114.9862, 114.9552, 115.1467,
    115.1318, 115.2325, 115.0671, 115.0337, 115.0233, 114.9526, 114.9999,
    115.0642, 115.1865, 115.2264, 115.5537, 115.5545, 115.7994, 115.7226,
    115.6986, 115.8065, 115.7342, 116.1176, 115.9290, 116.0082, 116.1508,
    116.5075, 116.5701, 116.3572, 116.2777, 116.3980, 116.4200, 116.6016,
    116.7802],
    "historical_context_text": [
      "Government bond yields alternated between firming and declining, shifting
      demand for higheryield assets.",
      (...)
      "Unexpected inflation readings lifted demand for inflationprotected assets
      and reshaped expectations for future price growth."
    ],
    "future_outlook_text": [
      "If cumulative policy guidance turns relatively more restrictive and
      safeasset yields persistently rise, sustained crossborder flows into dollar
      assets and tighter funding conditions will bolster demand for the dollar and
      lift the broad index.",
      "Should risk appetite recover and liquidity strains ease, persistent capital
      flows into higheryielding cyclical assets and a narrowing yield advantage
      will reduce dollar demand and weigh on the broad index."
    ],
    "prediction_horizon": 30,
    "options": ["rise", "unchanged", "fall"]
  },
  "answer": {
    "future_timeseries": [117.2434, 117.0456, 117.4010, 117.2417, 117.4048,
    117.3686, 117.6573, 116.8148, 116.4958, 116.7799, 116.7913, 116.7132,
    117.1927, 117.9082, 118.2564, 120.4945, 120.4439, 120.9417, 122.4875,
    124.1693, 125.0662, 124.9425, 126.1342, 125.5092, 124.7995, 122.4384,
    122.4097, 123.2997, 122.5394, 123.8033],
```

```
        "trend": "rise"
    }
}
```

Table 23: Illustrative data sample for the Text-Guided Counterfactual Forecasting task.

```
{
    "domain": "Economy",
    "task": "Text Guided Counterfactual Forecasting",
    "description": {
        "task_description": "The task is to classify the target variable trend 5
        steps ahead compared to the last data point as one of: rise, unchanged, or
        fall.",
        "data_description": "The following data is from the economy domain and
        contains the U.S. dollar broad index (DTWEXBGS). Daily variation is
        important, as short-term shocks often arise from economic releases, monetary
        policy expectations, and geopolitical events, while structural drivers such
        as trade flows and capital markets shape baseline conditions."
    },
    "data": {
        "history_timeseries": [120.2628, 120.2102, 120.2175, 120.1906, 120.0893,
        119.8069, 119.6781, 119.8890, 119.4641, 119.0646, 118.8447, 118.7168,
        119.2458, 119.0438, 119.0740, 119.4584, 119.4123, 119.2350, 119.6759,
        119.8659, 119.7118, 119.5618, 119.6971, 120.1579, 119.4293, 119.3179,
        119.0891, 118.0104, 117.5569, 117.4209],
        "historical_context_text": [
            "Movements in government bond yields altered interestrate differentials and
            influenced currency demand.",
            (...)
            "Heightened geopolitical tensions increased demand for safehaven currencies
            and pressured riskier assets."
        ],
        "future_outlook_text": [
            "If domestic data surprise to the downside and short-term yields fall,
            funding conditions loosen and risk-seeking plus carry flows reduce dollar
            demand."
        ],
        "prediction_horizon": 5,
        "options": ["rise", "unchanged", "fall"]
    },
    "answer": {
        "trend": [
            "fall",
            "unchanged"
        ]
    }
}
```

# C ADDITIONAL RESULTS AND ANALYSIS

## C.1 DOMAIN-WISE PERFORMANCE

### C.1.1 POLITICS DOMAIN

The tables below report the performance of various models on the Politics domain of the WIT benchmark. As shown, in Text-guided Short Term Forecasting, smaller models often achieved lower MSE when provided only with the historical time series. However, as model size increased, the combination of historical time series, historical context, and future outlook consistently yielded the best MSE performance.

In terms of accuracy (Acc), the full input combination (History_TS + History_CTX + Future_OUT) produced the highest performance across majority of models, irrespective of size. This observation highlights an important nuance: a lower MSE does not necessarily correspond to better forecasting quality in practical terms. In other words, a model that plays it safe may achieve low MSE but still have low accuracy, meaning it often predicts the wrong direction, which does not reflect good forecasting in real-world scenarios.

Overall, these results emphasize that evaluating forecasting performance requires multiple metrics. Solely relying on MSE may be misleading, especially when considering models of different scales and the influence of contextual information. Accuracy, together with MSE, provides a more comprehensive understanding of model behavior in text-guided TSF tasks.

Table 24: Full results of Text Guided Short Term Forecasting in Politics Domain

| Model | Short Term (Acc) | | | |
|---|---|---|---|---|
| | History_TS | History_TS +History_CTX | History_TS +Future_OUT | History_TS +History_CTX +Future_OUT |
| Mistral-7B-Instruct | 0.396 ± 0.049 | 0.466 ± 0.009 | 0.380 ± 0.045 | **0.478 ± 0.003** |
| Qwen2.5-7B-Instruct | 0.414 ± 0.009 | 0.453 ± 0.002 | 0.861 ± 0.001 | **0.890 ± 0.003** |
| gemma-3-27b-Instruct (4-bit) | 0.405 ± 0.005 | 0.442 ± 0.003 | **0.869 ± 0.002** | 0.864 ± 0.001 |
| Qwen3-32B (4-bit) | 0.391 ± 0.011 | 0.413 ± 0.014 | 0.865 ± 0.004 | **0.869 ± 0.004** |
| GPT-4o | 0.350 ± 0.009 | 0.396 ± 0.008 | 0.869 ± 0.002 | **0.919 ± 0.007** |

| Model | Short Term (MSE) | | | |
|---|---|---|---|---|
| | History_TS | History_TS +History_CTX | History_TS +Future_OUT | History_TS +History_CTX +Future_OUT |
| Mistral-7B-Instruct | **24.752 ± 1.395** | 34.644 ± 1.164 | 46.491 ± 11.137 | 42.937 ± 3.801 |
| Qwen2.5-7B-Instruct | **21.458 ± 0.268** | 25.975 ± 0.307 | 24.073 ± 1.359 | 29.067 ± 1.224 |
| gemma-3-27b-Instruct (4-bit) | **18.433 ± 0.041** | 18.945 ± 0.598 | 25.095 ± 0.310 | 20.824 ± 0.252 |
| Qwen3-32B (4-bit) | 25.731 ± 0.362 | 36.606 ± 5.540 | 33.702 ± 2.082 | **22.753 ± 0.072** |
| GPT-4o | 21.429 ± 0.257 | 21.564 ± 1.100 | 13.574 ± 0.554 | **13.494 ± 0.433** |

Table 25: Full results of Text Guided Long Term Forecasting in Politics Domain

| Model | Long Term (Acc) | | | |
|---|---|---|---|---|
| | History_TS | History_TS +History_CTX | History_TS +Future_OUT | History_TS +History_CTX +Future_OUT |
| Mistral-7B-Instruct | 0.411 ± 0.054 | 0.496 ± 0.009 | **0.585 ± 0.023** | 0.532 ± 0.017 |
| Qwen2.5-7B-Instruct | 0.418 ± 0.005 | 0.456 ± 0.002 | 0.681 ± 0.008 | **0.693 ± 0.004** |
| gemma-3-27b-Instruct (4-bit) | 0.497 ± 0.003 | 0.501 ± 0.002 | **0.710 ± 0.001** | 0.675 ± 0.001 |
| Qwen3-32B (4-bit) | 0.411 ± 0.001 | 0.428 ± 0.006 | **0.716 ± 0.005** | 0.685 ± 0.005 |
| GPT-4o | 0.384 ± 0.011 | 0.437 ± 0.003 | 0.630 ± 0.009 | **0.645 ± 0.005** |

Table 26: Full results of Text Guided Counterfactual Forecasting in Politics Domain

| Model | Counterfactual (Acc) | |
|---|---|---|
| | History_TS +Future_OUT | History_TS +History_CTX +Future_OUT |
| Mistral-7B-Instruct | 0.380 ± 0.039 | **0.419 ± 0.008** |
| Qwen2.5-7B-Instruct | 0.874 ± 0.002 | **0.896 ± 0.002** |
| gemma-3-27b-Instruct (4-bit) | **0.882 ± 0.000** | 0.867 ± 0.001 |
| Qwen3-32B (4-bit) | **0.934 ± 0.003** | 0.909 ± 0.006 |
| GPT-4o | 0.962 ± 0.002 | **0.969 ± 0.003** |

### C.1.2 SOCIETY DOMAIN

The tables below report the performance of various models on the Society domain of the WIT benchmark. As shown, in Text-guided Short Term Forecasting, smaller models often achieved lower MSE when provided only with the historical time series. However, as model size increased, the combination of historical time series, historical context, and future outlook consistently yielded the best MSE performance.

In terms of accuracy, the inclusion of future outlook led to the highest performance in both Short Term and Long Term Forecasting across the majority of models, regardless of size. This highlights the importance of future outlook information in Text-guided TSF.

Table 27: Full results of Text Guided Short Term Forecasting in Society Domain

| Model | Short Term (Acc) | | | |
|---|---|---|---|---|
| | History_TS | History_TS +History_CTX | History_TS +Future_OUT | History_TS +History_CTX +Future_OUT |
| Mistral-7B-Instruct | 0.600 ± 0.038 | **0.644 ± 0.008** | 0.457 ± 0.025 | 0.636 ± 0.010 |
| Qwen2.5-7B-Instruct | 0.633 ± 0.006 | 0.615 ± 0.006 | 0.984 ± 0.002 | **0.993 ± 0.000** |
| gemma-3-27b-Instruct (4-bit) | 0.661 ± 0.003 | 0.635 ± 0.002 | **0.998 ± 0.000** | 0.990 ± 0.001 |
| Qwen3-32B (4-bit) | 0.685 ± 0.007 | 0.685 ± 0.004 | **0.993 ± 0.000** | **0.993 ± 0.001** |
| GPT-4o | 0.667 ± 0.006 | 0.668 ± 0.002 | **0.994 ± 0.001** | 0.990 ± 0.002 |

| Model | Short Term (MSE) | | | |
|---|---|---|---|---|
| | History_TS | History_TS +History_CTX | History_TS +Future_OUT | History_TS +History_CTX +Future_OUT |
| Mistral-7B-Instruct | **32.377 ± 16.189** | 48.967 ± 4.871 | 41.925 ± 6.170 | 50.676 ± 8.077 |
| Qwen2.5-7B-Instruct | **8.341 ± 0.081** | 14.177 ± 1.544 | 10.713 ± 0.429 | 19.521 ± 2.991 |
| gemma-3-27b-Instruct (4-bit) | 9.534 ± 0.164 | 8.826 ± 0.084 | 7.280 ± 0.073 | **6.838 ± 0.065** |
| Qwen3-32B (4-bit) | 9.236 ± 0.189 | 10.677 ± 0.747 | 12.598 ± 2.903 | **7.405 ± 0.681** |
| GPT-4o | 8.489 ± 0.050 | 7.408 ± 0.199 | 5.471 ± 0.137 | **4.068 ± 0.206** |

Table 28: Full results of Text Guided Long Term Forecasting in Society Domain

| Model | Long Term (Acc) | | | |
|---|---|---|---|---|
| | History_TS | History_TS +History_CTX | History_TS +Future_OUT | History_TS +History_CTX +Future_OUT |
| Mistral-7B-Instruct | 0.595 ± 0.052 | 0.617 ± 0.002 | 0.796 ± 0.024 | **0.817 ± 0.010** |
| Qwen2.5-7B-Instruct | 0.688 ± 0.004 | 0.638 ± 0.003 | **0.885 ± 0.003** | 0.875 ± 0.002 |
| gemma-3-27b-Instruct (4-bit) | 0.694 ± 0.003 | 0.645 ± 0.005 | **0.892 ± 0.001** | 0.871 ± 0.001 |
| Qwen3-32B (4-bit) | 0.735 ± 0.001 | 0.723 ± 0.003 | **0.907 ± 0.001** | 0.890 ± 0.004 |
| GPT-4o | 0.756 ± 0.006 | 0.711 ± 0.004 | **0.866 ± 0.003** | 0.832 ± 0.003 |

Table 29: Full results of Text Guided Counterfactual Forecasting in Society Domain

| Model | Counterfactual (Acc) | |
|---|---|---|
| | History_TS +Future_OUT | History_TS +History_CTX +Future_OUT |
| Mistral-7B-Instruct | 0.478 ± 0.047 | **0.494 ± 0.008** |
| Qwen2.5-7B-Instruct | **0.945 ± 0.001** | 0.944 ± 0.001 |
| gemma-3-27b-Instruct (4-bit) | 0.949 ± 0.000 | **0.969 ± 0.000** |
| Qwen3-32B (4-bit) | 0.956 ± 0.001 | **0.970 ± 0.001** |
| GPT-4o | **0.990 ± 0.002** | 0.983 ± 0.005 |

### C.1.3 ENERGY DOMAIN

The tables below report the performance of various models on the Energy domain of the WIT benchmark. Unlike the Politics and Society domains, the prediction horizons in Energy domain are greater than 1, requiring the models to predict multiple consecutive time series steps. This makes the forecasting task particularly challenging. Consequently, for the Short Term Forecasting task, we report two MSE metrics: the average MSE across the full prediction horizon and the MSE on the last data point.

Overall, the trends across the two MSE metrics were similar. For all models except GPT, the lowest MSE was achieved when only the historical time series was provided. In terms of accuracy, the highest performance on the final predicted data point was observed when future outlook information was included, highlighting its importance for directional prediction in text-guided TSF, even in challenging continuous prediction settings.

Table 30: Full results of Text Guided Short Term Forecasting in Energy Domain

| Model | Short Term (Acc) | | | |
| --- | --- | --- | --- | --- |
| | History_TS | History_TS +History_CTX | History_TS +Future_OUT | History_TS +History_CTX +Future_OUT |
| Mistral-7B-Instruct | 0.374 ± 0.022 | 0.436 ± 0.029 | 0.458 ± 0.050 | **0.511 ± 0.023** |
| Qwen2.5-7B-Instruct | 0.441 ± 0.002 | 0.412 ± 0.009 | **0.633 ± 0.001** | 0.549 ± 0.004 |
| gemma-3-27b-Instruct (4-bit) | 0.488 ± 0.002 | 0.478 ± 0.003 | **0.633 ± 0.000** | 0.626 ± 0.001 |
| Qwen3-32B (4-bit) | 0.468 ± 0.005 | 0.465 ± 0.002 | **0.644 ± 0.002** | 0.621 ± 0.011 |
| GPT-4o | 0.500 ± 0.015 | 0.466 ± 0.013 | **0.630 ± 0.001** | 0.584 ± 0.018 |
| Model | Short Term - Full (MSE) | | | |
| | History_TS | History_TS +History_CTX | History_TS +Future_OUT | History_TS +History_CTX +Future_OUT |
| Mistral-7B-Instruct | **0.739 ± 0.166** | 1.413 ± 0.148 | 2.684 ± 0.530 | 1.512 ± 0.229 |
| Qwen2.5-7B-Instruct | **0.364 ± 0.013** | 0.399 ± 0.005 | 0.436 ± 0.011 | 0.421 ± 0.004 |
| gemma-3-27b-Instruct (4-bit) | **0.550 ± 0.066** | 19.439 ± 18.776 | 0.652 ± 0.070 | 0.736 ± 0.003 |
| Qwen3-32B (4-bit) | **0.502 ± 0.134** | 0.858 ± 0.072 | 1.270 ± 0.049 | 1.058 ± 0.061 |
| GPT-4o | 0.729 ± 0.390 | **0.610 ± 0.119** | 0.820 ± 0.117 | 0.832 ± 0.051 |
| Model | Short Term - Last Data Point (MSE) | | | |
| | History_TS | History_TS +History_CTX | History_TS +Future_OUT | History_TS +History_CTX +Future_OUT |
| Mistral-7B-Instruct | **0.601 ± 0.266** | 2.109 ± 0.646 | 7.547 ± 3.351 | 2.372 ± 0.618 |
| Qwen2.5-7B-Instruct | **0.271 ± 0.013** | 0.287 ± 0.011 | 0.320 ± 0.012 | 0.290 ± 0.000 |
| gemma-3-27b-Instruct (4-bit) | **0.363 ± 0.062** | 0.517 ± 0.017 | 0.460 ± 0.074 | 0.632 ± 0.009 |
| Qwen3-32B (4-bit) | **0.411 ± 0.150** | 0.890 ± 0.146 | 1.128 ± 0.063 | 1.030 ± 0.045 |
| GPT-4o | 0.862 ± 0.592 | **0.547 ± 0.150** | 0.838 ± 0.189 | 0.825 ± 0.072 |

Table 31: Full results of Text Guided Long Term Forecasting in Energy Domain

| Model | Long Term (Acc) | | | |
| --- | --- | --- | --- | --- |
| | History_TS | History_TS +History_CTX | History_TS +Future_OUT | History_TS +History_CTX +Future_OUT |
| Mistral-7B-Instruct | 0.429 ± 0.031 | 0.434 ± 0.025 | **0.608 ± 0.033** | 0.552 ± 0.025 |
| Qwen2.5-7B-Instruct | 0.451 ± 0.010 | 0.435 ± 0.010 | **0.937 ± 0.002** | 0.838 ± 0.008 |
| gemma-3-27b-Instruct (4-bit) | 0.482 ± 0.006 | 0.486 ± 0.003 | **0.943 ± 0.001** | 0.939 ± 0.002 |
| Qwen3-32B (4-bit) | 0.474 ± 0.006 | 0.472 ± 0.006 | **0.940 ± 0.002** | 0.881 ± 0.009 |
| GPT-4o | 0.506 ± 0.004 | 0.470 ± 0.007 | **0.940 ± 0.003** | 0.919 ± 0.009 |

Table 32: Full results of Text Guided Counterfactual Forecasting in Energy Domain

| Model | Counterfactual (Acc) | |
| --- | --- | --- |
| | History_TS +Future_OUT | History_TS +History_CTX +Future_OUT |
| Mistral-7B-Instruct | 0.468 ± 0.025 | **0.594 ± 0.004** |
| Qwen2.5-7B-Instruct | 0.681 ± 0.004 | **0.773 ± 0.005** |
| gemma-3-27b-Instruct (4-bit) | **0.647 ± 0.000** | 0.639 ± 0.001 |
| Qwen3-32B (4-bit) | 0.649 ± 0.003 | **0.658 ± 0.009** |
| GPT-4o | 0.679 ± 0.001 | **0.696 ± 0.009** |

### C.1.4 ECONOMY DOMAIN

For the Economy domain, the prediction horizon similarly spans more than one time step. Accordingly, MSE was reported both as the average over the full prediction horizon and at the final predicted data point. The observed trends mirrored those in the Energy domain: models generally achieved the lowest MSE when only historical time series were provided, whereas accuracy was highest when future outlook information was included, underscoring its importance for directional prediction in text-guided TSF.

Table 33: Full results of Text Guided Short Term Forecasting in Economy Domain

| Model | Short Term (Acc) | | | |
| --- | --- | --- | --- | --- |
| | History_TS | History_TS +History_CTX | History_TS +Future_OUT | History_TS +History_CTX +Future_OUT |
| Mistral-7B-Instruct | 0.392 ± 0.029 | 0.445 ± 0.014 | 0.485 ± 0.037 | **0.517 ± 0.011** |
| Qwen2.5-7B-Instruct | 0.517 ± 0.006 | 0.529 ± 0.007 | 0.610 ± 0.001 | **0.641 ± 0.003** |
| gemma-3-27b-Instruct (4-bit) | 0.461 ± 0.002 | 0.512 ± 0.003 | 0.646 ± 0.000 | **0.653 ± 0.002** |
| Qwen3-32B (4-bit) | 0.504 ± 0.005 | 0.513 ± 0.004 | **0.642 ± 0.002** | 0.631 ± 0.004 |
| GPT-4o | 0.510 ± 0.008 | 0.490 ± 0.009 | **0.646 ± 0.000** | 0.643 ± 0.001 |

| Model | Short Term - Full (MSE) | | | |
| --- | --- | --- | --- | --- |
| | History_TS | History_TS +History_CTX | History_TS +Future_OUT | History_TS +History_CTX +Future_OUT |
| Mistral-7B-Instruct | 4.853 ± 4.205 | **0.824 ± 0.028** | 12.451 ± 11.364 | 1.494 ± 0.038 |
| Qwen2.5-7B-Instruct | **0.552 ± 0.015** | 0.695 ± 0.020 | 0.892 ± 0.009 | 0.789 ± 0.048 |
| gemma-3-27b-Instruct (4-bit) | **0.598 ± 0.005** | 0.644 ± 0.004 | 0.646 ± 0.013 | 0.744 ± 0.012 |
| Qwen3-32B (4-bit) | **0.541 ± 0.009** | 0.551 ± 0.014 | 1.043 ± 0.003 | 0.874 ± 0.016 |
| GPT-4o | **0.491 ± 0.012** | 0.515 ± 0.016 | 0.704 ± 0.043 | 0.597 ± 0.022 |

| Model | Short Term - Last Data Point (MSE) | | | |
| --- | --- | --- | --- | --- |
| | History_TS | History_TS +History_CTX | History_TS +Future_OUT | History_TS +History_CTX +Future_OUT |
| Mistral-7B-Instruct | 26.541 ± 13.294 | **1.250 ± 0.036** | 1.797 ± 0.083 | 2.304 ± 0.073 |
| Qwen2.5-7B-Instruct | **0.968 ± 0.015** | 1.071 ± 0.023 | 1.346 ± 0.002 | 1.169 ± 0.046 |
| gemma-3-27b-Instruct (4-bit) | **1.133 ± 0.014** | 1.172 ± 0.006 | 1.242 ± 0.024 | 1.361 ± 0.021 |
| Qwen3-32B (4-bit) | **0.924 ± 0.021** | 0.968 ± 0.009 | 2.108 ± 0.009 | 1.757 ± 0.010 |
| GPT-4o | **0.875 ± 0.026** | 0.920 ± 0.042 | 1.368 ± 0.079 | 1.149 ± 0.048 |

Table 34: Full results of Text Guided Long Term Forecasting in Economy Domain

| Model | Long Term (Acc) | | | |
|---|---|---|---|---|
| | `History_TS` | `History_TS` `+History_CTX` | `History_TS` `+Future_OUT` | `History_TS` `+History_CTX` `+Future_OUT` |
| Mistral-7B-Instruct | 0.443 ± 0.018 | 0.444 ± 0.018 | 0.469 ± 0.023 | **0.499 ± 0.024** |
| Qwen2.5-7B-Instruct | 0.452 ± 0.012 | 0.459 ± 0.005 | **0.570 ± 0.003** | 0.531 ± 0.013 |
| gemma-3-27b-Instruct (4-bit) | 0.476 ± 0.003 | 0.492 ± 0.002 | **0.567 ± 0.001** | 0.555 ± 0.005 |
| Qwen3-32B (4-bit) | 0.469 ± 0.021 | 0.473 ± 0.014 | **0.567 ± 0.003** | 0.536 ± 0.013 |
| GPT-4o | 0.490 ± 0.003 | 0.495 ± 0.024 | **0.555 ± 0.009** | 0.543 ± 0.012 |

Table 35: Full results of Text Guided Counterfactual Forecasting in Economy Domain

| Model | Counterfactual (Acc) | |
|---|---|---|
| | `History_TS` `+Future_OUT` | `History_TS` `+History_CTX` `+Future_OUT` |
| Mistral-7B-Instruct | 0.507 ± 0.023 | **0.539 ± 0.006** |
| Qwen2.5-7B-Instruct | 0.525 ± 0.009 | **0.574 ± 0.005** |
| gemma-3-27b-Instruct (4-bit) | 0.597 ± 0.001 | **0.604 ± 0.001** |
| Qwen3-32B (4-bit) | 0.536 ± 0.006 | **0.588 ± 0.002** |
| GPT-4o | 0.577 ± 0.004 | **0.593 ± 0.003** |

## C.2 MORE ON HISTORICAL CONTEXT ABLATION

### C.2.1 DETAILS ON EXPERIMENT

We conducted experiments on the Politics domain of the WIT benchmark for a representative LLM, comparing the performance of different historical context selection strategies. For the LLM-filter strategy, the following prompt was used: `"Select exactly the 4 most important events from the list. Do not provide explanations. Only list the 4 events:"`, whereas for the LLM-summary strategy, the prompt was: `"Summarize only the most important historical events from the following list. Be concise and start directly with the summary:"`. Prompt engineering was kept minimal to focus on evaluating the strategies themselves.

## C.2.2 RESULTS

Table 36: Full results of Short Term Forecast accuracy on the Politics domain of the WIT benchmark for a representative LLM, comparing different historical context selection strategies. For each model and input configuration, the best-performing result is underlined.

| Model | Method | Short Term (Acc) | | | |
| | | History_TS | History_TS +History_CTX | History_TS +Future_OUT | History_TS +History_CTX +Future_OUT |
|---|---|---|---|---|---|
| Mistral-7B-Instruct | default | 0.494 | 0.483 | 0.469 | 0.483 |
| | recent4 | - | 0.462 | - | 0.483 |
| | random4 | - | 0.485 | - | 0.490 |
| | llm_filter | - | 0.483 | - | 0.478 |
| | llm_summary | - | 0.478 | - | 0.476 |
| Qwen2.5-7B-Instruct | default | 0.42 | 0.455 | 0.863 | 0.893 |
| | recent4 | - | 0.436 | - | 0.870 |
| | random4 | - | 0.455 | - | 0.872 |
| | llm_filter | - | 0.466 | - | 0.861 |
| | llm_summary | - | 0.422 | - | 0.875 |
| gemma-3-27b-Instruct (4-bit) | default | 0.415 | 0.436 | 0.868 | 0.863 |
| | recent4 | - | 0.408 | - | 0.856 |
| | random4 | - | 0.411 | - | 0.865 |
| | llm_filter | - | 0.446 | - | 0.859 |
| | llm_summary | - | 0.427 | - | 0.859 |
| Qwen3-32B (4-bit) | default | 0.376 | 0.415 | 0.868 | 0.865 |
| | recent4 | - | 0.404 | - | 0.863 |
| | random4 | - | 0.404 | - | 0.863 |
| | llm_filter | - | 0.441 | - | 0.861 |
| | llm_summary | - | 0.404 | - | 0.866 |
| GPT-4o | default | 0.355 | 0.392 | 0.872 | 0.921 |
| | recent4 | - | 0.399 | - | 0.900 |
| | random4 | - | 0.348 | - | 0.910 |
| | llm_filter | - | 0.411 | - | 0.900 |
| | llm_summary | - | 0.388 | - | 0.896 |

| Model | Method | Short Term (MSE) | | | |
| | | History_TS | History_TS +History_CTX | History_TS +Future_OUT | History_TS +History_CTX +Future_OUT |
|---|---|---|---|---|---|
| Mistral-7B-Instruct | default | 23.054 | 33.316 | 25.895 | 37.644 |
| | recent4 | - | 21.266 | - | 30.762 |
| | random4 | - | 26.013 | - | 41.133 |
| | llm_filter | - | 33.579 | - | 43.842 |
| | llm_summary | - | 30.544 | - | 45.395 |
| Qwen2.5-7B-Instruct | default | 21.119 | 25.370 | 22.269 | 27.821 |
| | recent4 | - | 21.452 | - | 24.574 |
| | random4 | - | 25.272 | - | 24.613 |
| | llm_filter | - | 26.432 | - | 26.207 |
| | llm_summary | - | 24.395 | - | 24.622 |
| gemma-3-27b-Instruct (4-bit) | default | 18.384 | 19.195 | 25.201 | 20.494 |
| | recent4 | - | 19.265 | - | 18.514 |
| | random4 | - | 19.487 | - | 22.202 |
| | llm_filter | - | 22.886 | - | 26.841 |
| | llm_summary | - | 19.625 | - | 19.305 |
| Qwen3-32B (4-bit) | default | 25.009 | 26.340 | 32.549 | 22.634 |
| | recent4 | - | 25.981 | - | 22.346 |
| | random4 | - | 24.103 | - | 24.587 |
| | llm_filter | - | 26.073 | - | 20.380 |
| | llm_summary | - | 24.811 | - | 21.454 |
| GPT-4o | default | 21.341 | 21.329 | 13.020 | 14.070 |
| | recent4 | - | 21.475 | - | 10.854 |
| | random4 | - | 19.930 | - | 12.222 |
| | llm_filter | - | 21.429 | - | 12.501 |
| | llm_summary | - | 22.051 | - | 11.645 |

Table 37: Full results of Long Term Forecast accuracy on the Politics domain of the WIT benchmark for a representative LLM, comparing different historical context selection strategies. For each model and input configuration, the best-performing result is underlined.

| Model | Method | Long Term (Acc) | | | |
|---|---|---|---|---|---|
| | | History_TS | History_TS +History_CTX | History_TS +Future_OUT | History_TS +History_CTX +Future_OUT |
| Mistral-7B-Instruct | default | 0.517 | 0.502 | 0.632 | 0.563 |
| | recent4 | - | 0.485 | - | 0.539 |
| | random4 | - | 0.517 | - | 0.539 |
| | llm_filter | - | 0.498 | - | 0.537 |
| | llm_summary | - | 0.507 | - | 0.534 |
| Qwen2.5-7B-Instruct | default | 0.417 | 0.451 | 0.695 | 0.693 |
| | recent4 | - | 0.415 | - | 0.707 |
| | random4 | - | 0.420 | - | 0.695 |
| | llm_filter | - | 0.442 | - | 0.700 |
| | llm_summary | - | 0.407 | - | 0.695 |
| Gemma-3-27b-Instruct (4-bit) | default | 0.495 | 0.498 | 0.712 | 0.676 |
| | recent4 | - | 0.449 | - | 0.698 |
| | random4 | - | 0.466 | - | 0.690 |
| | llm_filter | - | 0.485 | - | 0.678 |
| | llm_summary | - | 0.485 | - | 0.668 |
| Qwen3-32B (4-bit) | default | 0.412 | 0.439 | 0.717 | 0.695 |
| | recent4 | - | 0.420 | - | 0.678 |
| | random4 | - | 0.451 | - | 0.688 |
| | llm_filter | - | 0.454 | - | 0.681 |
| | llm_summary | - | 0.434 | - | 0.698 |
| GPT-4o | default | 0.400 | 0.437 | 0.629 | 0.644 |
| | recent4 | - | 0.434 | - | 0.642 |
| | random4 | - | 0.390 | - | 0.639 |
| | llm_filter | - | 0.459 | - | 0.629 |
| | llm_summary | - | 0.405 | - | 0.624 |

Table 38: Full results of Counterfactual Forecasting accuracy on the Politics domain of the WIT benchmark for a representative LLM, comparing different historical context selection strategies. For each model and input configuration, the best-performing result is underlined.

| Model | Method | Counterfactual (Acc) | |
|---|---|---|---|
| | | History_TS +Future_OUT | History_TS +History_CTX +Future_OUT |
| Mistral-7B-Instruct | default | 0.457 | 0.436 |
| | recent4 | - | 0.460 |
| | random4 | - | 0.454 |
| | llm_filter | - | 0.457 |
| | llm_summary | - | 0.441 |
| Qwen2.5-7B-Instruct | default | 0.874 | 0.895 |
| | recent4 | - | 0.901 |
| | random4 | - | 0.879 |
| | llm_filter | - | 0.887 |
| | llm_summary | - | 0.887 |
| gemma-3-27b-Instruct (4-bit) | default | 0.882 | 0.868 |
| | recent4 | - | 0.863 |
| | random4 | - | 0.882 |
| | llm_filter | - | 0.876 |
| | llm_summary | - | 0.874 |
| Qwen3-32B (4-bit) | default | 0.936 | 0.903 |
| | recent4 | - | 0.919 |
| | random4 | - | 0.922 |
| | llm_filter | - | 0.914 |
| | llm_summary | - | 0.930 |
| GPT-4o | default | 0.965 | 0.970 |
| | recent4 | - | 0.962 |
| | random4 | - | 0.954 |
| | llm_filter | - | 0.960 |
| | llm_summary | - | 0.952 |

## D  IMPLEMENTATION DETAILS

### D.1  EXPERIMENTAL SETTINGS

**Scenario-guided Multimodal Forecasting**  We ran general-purpose LLMs and multimodal LLM fine-tuned for time series (as denoted in FTS) on a single NVIDIA A6000 GPU with 48GB RAM. Mistral-7B-Instruct-v0.3 (Jiang et al., 2023), Qwen2.5-7B-Instruct (Qwen et al., 2025), Gemma-3-27B-IT (Team et al., 2025), Qwen3-32B (Yang et al., 2025), and Time-MQA (Qwen2.5-7B) (Kong et al., 2025a) were tested. For Gemma-3-27B-IT and Qwen3-32B, we adopted 4-bit quantization to operate within available computational resources. Inference for GPT-4o (OpenAI et al., 2024) was performed using the GPT API. All experiments were repeated across three random seeds to ensure robustness.

**Unimodal (Time Series) Forecasting**  As unimodal baselines, we also ran recent Transformer-based TSFMs on a single NVIDIA A6000 GPU with 48GB RAM. Chronos (Chronos-Bolt-Base) (Ansari et al., 2024), Moirai (Moirai-1.1-R-Large) (Woo et al., 2024), and TimesFM (TimesFM-2.5-200M) (Das et al., 2024) were all tested in a zero-shot setting.

For statistical methods, we implemented ARIMA (Box & Jenkins, 1976), ETS (state-space exponential smoothing) (Hyndman et al., 2008), and Holt–Winters classical exponential smoothing (Brown, 2004). All were applied in a univariate setting, with hyperparameters (e.g., ARIMA $(p, d, q)$ orders, ETS trend/seasonal/damping options, and Holt–Winters seasonality) selected automatically by grid search using Akaike Information Criterion (AIC). When model fitting failed or data were insufficient, forecasts defaulted to naive persistence (last value repeated).

### D.2  PROMPT TEMPLATES

Below are the prompt templates used in our experiments. Depending on the input combination, the corresponding template was applied. If an input configuration is not explicitly specified, all components-data description, historical time series, historical context, and future outlook—were included. Since LLMs are highly sensitive to prompt engineering, we deliberately kept prompt modifications minimal to isolate and assess the effectiveness and utility of our dataset itself.

#### D.2.1  ONLY TIME SERIES

Table 39: Prompt template used for text-guided TSF with time series data only.

```
You are a time-series forecasting expert.

{s['description']['task_description']}

Historical time series: {s['data']['history_timeseries']}

Do NOT provide any explanation or reasoning. Output only a single number.
```

#### D.2.2  DATA DESCRIPTION + TIME SERIES

Table 40: Prompt template used for text-guided TSF with data description and time series data.

```
You are a time-series forecasting expert.

{s['description']['data_description']}
{s['description']['task_description']}

Historical time series: {s['data']['history_timeseries']}

Do NOT provide any explanation or reasoning. Output only one of the provided options.
```

### D.2.3 DATA DESCRIPTION + TIME SERIES + HISTORICAL CONTEXT

Table 41: Prompt template used for text-guided TSF with data description, time series data, and historical context.

```
You are a time-series forecasting expert.

{s['description']['data_description']}
{s['description']['task_description']}

Historical time series: {s['data']['history_timeseries']}
Historical context: {chr(10).join(s['data']['historical_context_text'])}

Do NOT provide any explanation or reasoning. Output only one of the provided options.
```

### D.2.4 DATA DESCRIPTION + TIME SERIES + FUTURE OUTLOOK

Table 42: Prompt template used for text-guided TSF with data description, time series data, and future outlook.

```
You are a time-series forecasting expert.

{s['description']['data_description']}
{s['description']['task_description']}

Historical time series: {s['data']['history_timeseries']}

Future scenario: {chr(10).join(s['data']['future_outlook_text'])}

Do NOT provide any explanation or reasoning. Output only one of the provided options.
```

### D.2.5 DATA DESCRIPTION + TIME SERIES + HISTORICAL CONTEXT + FUTURE OUTLOOK

Table 43: Prompt template used for text-guided TSF with data description, time series data, historical context, and future outlook.

```
You are a time-series forecasting expert.

{s['description']['data_description']}
{s['description']['task_description']}

Historical time series: {s['data']['history_timeseries']}
Historical context: {chr(10).join(s['data']['historical_context_text'])}

Future scenario: {chr(10).join(s['data']['future_outlook_text'])}

Do NOT provide any explanation or reasoning. Output only one of the provided options.
```

## E    USE OF LARGE LANGUAGE MODELS (LLMS) IN PAPER WRITING

We used LLMs *only* to aid and polish writing (grammar, fluency, concision) and to suggest minor LaTeX phrasing/formatting; we did *not* use LLMs for retrieval and discovery (e.g., finding related work) or for research ideation. LLMs did not generate technical content or citations, and did not contribute at the level of a contributing author. All text and claims were authored, verified, and finalized by the authors, with LLM-suggested edits accepted only after manual review to avoid hallucinations or unsupported statements.

# F    ETHICS STATEMENT

The datasets introduced in this work are constructed entirely from publicly available and non-sensitive sources. All textual and numerical data were collected from reputable public institutions and media outlets with appropriate attribution. No personally identifiable information or private data were included. We anticipate that this benchmark will primarily benefit the research community by enabling more realistic and rigorous evaluation of multimodal forecasting methods, and we do not foresee direct risks of harm associated with its use.

