# OpenReview forum: "What If TSF: A Multimodal Benchmark for Conditional Time Series Forecasting with Plausible Scenarios"
_ICLR.cc/2026/Conference — ICLR 2026 Conference Withdrawn Submission_

### Official Review · Reviewer_atme · 2025-10-28

**Soundness:** 2
**Presentation:** 4
**Contribution:** 2
**Rating:** 4
**Confidence:** 4

**Summary:**

### *Summary*

- The authors present a novel benchmark for context-aided forecasting with what-if scenarios over 4 domains.  This benchmark distinguishes between static and dynamic context, and adds counterfactual future context. The authors measure using directional accuracy and MSE (no MSE for long-term forecasts). The authors evaluate a diverse subset of LLMs, TSFMs, Time-MQA and statistical methods (e.g. ARIMA).

  ### *Contributions*

- Novel benchmark for context-aided forecasting that distinguishes itself particularly by considering counterfactuals and evaluating directional accuracy.

**Strengths:**

Overall, the paper is well-structured, with evidence mostly supporting its claims (see weaknesses and questions).

### *Originality*

- Enlightening comparison against relevant previous work (Time-MMD and Context is Key (CiK), Table 1)., with introduction of interesting counterfactual scenarios.

  ### *Quality*

- Good set of baselines
- Diverse metrics

  ### *Clarity*

- Clear text

  ### *Significance*

- Important, timely topic

**Weaknesses:**

The main weaknesses of the paper lie in the validation of its original contributions.

### *Originality*

- The distinction from previous benchmarks (Time-MMD and Context is Key (CiK) ) is vague, and it’s unclear what analysis backs up table 1\.
  - While the work states that in practice, Time-MMD has redundancies and incomplete texts, the authors do not back these claims up with an analysis of the dataset. A better understanding of the limitations  of Time-MMD would certainly be useful to the community.
  - The discussion in 4.2 would be greatly improved by actual comparisons to examples from the WIT benchmark to ground the claimed improvements.
  - While 4.3 discusses three components (time series, historical context and future information), figure 4 of CiK shows that their classification of context subsumes this categorization. Improving 4.2 with concrete examples from WIT that highlight examples would fix this issue.

- Contemporaneous work that is highly related: https://openreview.net/forum?id=YRp4xqTs3n

  ### *Quality*

- If you measure using directional accuracy, how do you estimate overshooting or undershooting estimates? A decline in the dollar broad index of 10% vs 50% is a very different thing
- The analysis of why historical context does not improve accuracy is absent. Beyond the observed results, it would be important to understand *why* this is the case, especially since the benchmark itself is the main contribution of the article.
- An intact time series signal from before an LLM's knowledge cutoff can leak the future answer to an LLM that has been pre-trained on these time series, especially if the values are somewhat unique. How do you validate that your memorization mitigation strategy works?
- I’m not sure what the takeaway here is from the ablation, it’s unclear what this contributes to my understanding of the benchmark.

**Questions:**

### *change your opinion*

In my mind, this paper is currently a 5: I wouldn't mind it being accepted. However, I think that additional validation is required to ensure that this benchmark will be useful to the context-aided forecasting community in its current form. The main things I would like to see to up my score are an assessment of the relevance of the historical context, validation of the memorization mitigation. Beyond that, adding additional baselines and analyses of forecasting instances could improve my score, after a rigorous validation of the relevance of the context.

- In which cases is the historical context informative or uninformative? A human eval here would be useful to confirm whether the information is indeed relevant or not to the forecasting task, beyond it being merely logically compatible. Otherwise, what's the point of including it in the benchmark?

  ### *clarify a confusion*

- Where are the details of your human evaluation protocol for validating the relevance of the context (section 4.6?)

  ### *address a limitation*

- Have you considered estimating model abilities when context is less informative, e.g. removing the “likely to weaken significantly” part in the example of figure 1 to see if the model can deduce the impact of temperature on demand for natural gas itself?
- How do you ensure that the news is not actually updated after the fact? Many articles are first published, then continuously updated with relevant facts afterward, which may leak the answer.
- It’s not only the textual facts, but also the time series themselves which present potential leakage problems. LLMs are pretrained on time series data in the wild, even if it is input as text tokens. How do you prevent this form of leakage?

---

> ### Author Response · Authors · 2025-11-28
>
> We deeply appreciate the reviewer’s thoughtful and constructive feedback.
>
> In response to all reviewers’ comments, we plan to make the following revisions:
>
> - Revise Section 3 to more clearly explain the tasks.
> - Revise Section 4 to better position and distinguish our benchmark dataset from existing benchmarks, and to more explicitly describe our data construction and validation pipeline.
> - Revise Section 5 by incorporating extended experiments and elevating the reviewer-suggested analysis (estimating model abilities when the provided context is less informative) to our main ablation study.
>
> Below, we summarize our planned revisions with respect to the weaknesses and questions you raised.
>
> Due to time constraints during the rebuttal period, it may not be feasible to implement all of these changes directly in the next version of the manuscript.
>
> Nevertheless, we have formulated a detailed revision plan based on your feedback, and we would be very grateful if you could comment on whether this plan is coherent and sufficiently systematic.
>
> ---
>
> ### **1. More specific comparison with existing datasets and clarification of our benchmark’s contribution**
>
> - For multimodal forecasting, the most representative existing benchmarks are *Time-MMD* and *CiK*.
> - However, they have several limitations that make them less suitable as practical evaluation platforms. Below, we summarize concrete issues.
>     - We will also illustrate with examples and an updated Notes column in Table 1 in the revised manuscript.
>
> ### a. Time-MMD
> **(1) The textual data in Time-MMD are often incompletely preprocessed, so additional cleaning is required before use, and there exist many intervals where only the time series are available with no meaningful text.**
>
> - Several textual entries are essentially placeholders or carry almost no task-relevant information, for example:
>     - Generic template-like descriptions such as *“Objective facts about the Monthly Retail broiler Composite situation. No direct information about prices is provided in the report.”* (Agriculture / report).
>     - Numerous segments with NA, blank strings, or very sparse text in domains such as Health AFR, Health US, Security (report), or Climate (search).
>     - Mismatched entries like *“Objective facts about the influenza situation: NA”* attached to domains such as Traffic or SocialGood.
> - The format and granularity of the text are not standardized across time and domains, for example:
>     - Some Climate / report entries are short, single-sentence summaries, while others are long multi-sentence paragraphs describing multiple regions and periods in a highly heterogeneous way.
>         - *(2024.02.01~29) The contiguous U.S. experienced a drier-than-normal month in February 2024, with 23% of the area falling in the moderate to extreme drought categories based on the Palmer Drought Index.*
>         - *(2024.04.01~30) Objective facts about the Monthly Contiguous U.S. Precipitation situation. April 2024 was wetter than normal … The above-normal precipitation in the northern Plains to Midwest, …*
>     - This leads to substantial variation in prompt length and structure, which makes it difficult to isolate the effect of textual context on forecasting performance.
>
>  **Our benchmark (contribution):**
> - We systematically remove such incomplete or low-quality cases through a combination of LLM-based preprocessing and human expert verification.
> - All textual descriptions are normalized to a consistent format, and we ensure that each forecast instance is paired with non-empty, semantically meaningful textual context.
>     - Differences in model performance can be more reliably attributed to the informativeness of the context rather than artifacts of noisy preprocessing.
>
> **(2) In addition, there are many cases where the textual descriptions are redundant or temporally misaligned with the associated time series.**
>
> - (energy / report) A text snippet for the 2011.12.19~23 states: *“The national average retail regular gasoline price increased to $3.258 per gallon on December 26, 2011, 0.029 dollar per gallon more than last week ...”*
> - In this setting, the time series already provides numerical values for regional retail gasoline prices, and the target variable is the national average. For the 2011.12.19~23, only the 2011.12.19 time-series point is present, followed by the 2011.12.26 point.
>     - The textual information for the 2011.12.19~23 refers to the price on 2011.12.26 is temporally misaligned.
>     - Information such as “how much the price increased compared to last week” is already directly encoded in the time series, making the text largely redundant with respect to the forecasting task.
>
> **Our benchmark (contribution):**
>
> - Using a combination of LLM-based preprocessing and human expert verification, we systematically remove redundant textual summaries and temporally misaligned descriptions for each domain.
>     - The remaining context is genuinely complementary to the time series.

---

> ### Author Response · Authors · 2025-11-28
> **Official Comment by Authors (Cont')**
>
> ### b. CiK
>
> **(1) Many contextual descriptions remain highly specific and not fully de-identified.**
>
> - The context often contains detailed real-world entities and descriptions in a raw form, which raises concerns that future, more capable LLMs may simply memorize the benchmark rather than genuinely learning to reason from the context.
> - For example, a MontrealFire task includes text such as:
>     - *“The Montreal Fire Department is in charge of responding to various kinds of public safety incidents. This is the number of field fire incidents responded to by Montreal firefighters in the borough of Mercier-Hochelaga-Maisonneuve.”*
> - Such highly specific descriptions make the benchmark less robust to memorization and may provide overly direct cues tied to particular locations or agencies.
>
> **Our benchmark (contribution):**
>
> - During preprocessing, we explicitly de-identify concrete entities and locations in the context so that models cannot rely on trivial memorization of specific phrases or named entities.
> - Instead, the textual information is preserved at a level that remains semantically rich while reducing the risk of direct lookup-style memorization.
>
> **(2) Future information is often specified in an overly precise and unrealistic manner.**
>
> - In real-world applications, future conditions are rarely known with exact numerical precision. However, in CiK, future scenarios are frequently framed as strict quantitative constraints that guide intervention-based forecasting.
> - For instance, in the DecreaseInTrafficInPrediction task, the context states:
>     - *“Suppose that there is an accident on the road and there is 20.0% of the usual traffic from 2024-01-18 14:00:00 for 5 hours.”*
> - Such descriptions assume knowledge of highly specific future magnitudes and durations that would not be realistically available at prediction time
> - They effectively constrain the target trajectory in a way that may not reflect practical forecasting settings.
>
> **Our benchmark (contribution):**
>
> - Instead of providing exact numerical futures, our benchmark offers high-level scenario-style descriptions of potential future events (e.g., qualitative changes).
> - These scenarios are designed to guide the forecast direction without revealing fully specified future values, thereby better aligning with realistic decision-making contexts.
>
> ---
>
> ### **2. Existence of concurrent work**
>
> - The concurrent work mentioned by the reviewer refers to a paper that was submitted to the same ICLR cycle.
> - That paper primarily proposes a new modeling framework, whereas our goal is to develop a robust benchmark that can *support and evaluate* such frameworks.
> - In this sense, we view our contribution as complementary rather than competing: our benchmark is designed to provide a standardized, realistic testbed on which framework-level advances like the concurrent work can be systematically assessed.
>
>
> ---
>
> ### **3. Concern about the evaluation metric (3-way directional accuracy)**
>
> **On overshooting vs. undershooting:**
>
> - We acknowledge that overshooting and undershooting are distinct error modes. Initially, we considered a more fine-grained classification scheme (beyond 3-way) to explicitly differentiate them.
> - However, from a counterfactual-generation perspective, defining clean, interpretable counterfactual labels becomes increasingly ambiguous as the number of classes grows.
> - It is often unclear where to draw quantitative thresholds that meaningfully separate “slightly” vs. “strongly” over- or under-shooting.
>
> **Empirical difficulties with magnitude-sensitive labels for LLMs:**
>
> - In early experiments during benchmark construction, we tested a 5-way classification setup (e.g., surge, rise, unchanged, fall, crash) for LLM-based forecasters.
> - We observed that LLMs struggled to reliably understand and use the corresponding magnitude scales, often collapsing to a single preferred label rather than making calibrated distinctions.
> - We similarly found that prompting LLMs to predict continuous error magnitudes or MSE-related quantities led to unstable and poorly calibrated behavior, suggesting that LLMs have difficulty handling precise numerical scales in this setting.
>
>
>
> **Rationale for our 3-way directional metric and future extensions:**
>
> - Given these challenges, we adopted a 3-way directional accuracy metric as a pragmatic compromise that is both robust to scale-misinterpretation and practically evaluable for current LLMs.
> - At the same time, we fully agree that magnitude-sensitive metrics are important.
> - Where feasible, we plan to incorporate additional magnitude-aware evaluation criteria in future extensions of the benchmark, once we identify formulations that remain stable and interpretable for LLM-based models.

---

> ### Author Response · Authors · 2025-11-28
> **Official Comment by Authors (Cont')**
>
> ### **4. Need for additional validation to demonstrate the benchmark’s usefulness for context-aware forecasting**
>
> **Assessing the relevance of historical context**
>
> - During data construction, we already verified contextual relevance using LLM-based filtering (e.g., Δ-based checks) and aligned the textual information only when the model judged it relevant.
> - However, because the generation pipeline did not output an explicit quantitative relevance score, we plan to provide additional post-hoc validation in the revised manuscript.
> - Our post-hoc validation will include:
>     - **Human expert evaluation**
>         - Domain experts (with at least a bachelor’s-level background in the relevant field) will manually examine sample pairs via a structured Google Form and assess whether the provided context is meaningfully relevant to the target series.
>     - **Representation-level validation using pretrained foundation models**
>         - For each sample, we will extract time-series embeddings using a frozen TSFM encoder (e.g., TimesFM-2.5) and textual embeddings using a frozen general-purpose LLM encoder.
>         - Using HSIC and related dependence measures, we will examine how strongly the pretrained representations of the two modalities are statistically associated.
>
> **When is historical context informative or uninformative? (Why might relevance not translate into forecasting gains?)**
>
> - We acknowledge that the current manuscript lacks analysis explaining why historical context sometimes fails to improve multimodal forecast accuracy, even when the context is judged relevant.
> - To address this, our additional validation will include:
>     - **Human expert prediction task**
>         - Experts will be asked to predict directional changes directly from each sample’s context to quantify how informative the text is for humans.
>     - **Analysis of practically uninformative but relevant context**
>         - Even when historical text is semantically relevant, it may still have limited forecasting value.
>         - For example, when it lacks leading indicators or provides only descriptive rather than predictive cues.
>         - We will conduct a detailed post-hoc investigation of such cases once all annotations are collected.
>
> **Validation of memorization mitigation (LLM leakage)**
>
> - **Knowledge-cutoff–based analysis**
>     - For the Mistral-7B-Instruct-v0.3 model, we estimate the knowledge cutoff to be around April 2023.
>     - For instance, we sample 50 instances before and 50 instances after the cutoff date (Politics domain, Task 1) and compare performance.
>     - The results show no significant performance difference between pre- and post-cutoff data; if anything, directional accuracy is slightly higher on post-cutoff examples.
>     - **A similar post-hoc analysis using only raw time series (without textual context)** also reveals no advantage on pre-cutoff data, and in fact shows better performance on post-cutoff data, suggesting that direct memorization from prior web exposure is unlikely.
>
> *Short Term Forecasting for Politics Domain (Task 1)*
> |Category|MSE(history_ts)|MSE(history_ts+historical & future outlook text)|Acc(history_ts)|Acc(history_ts+historical & future outlook text)|
> |---|---|---|---|---|
> |Before(random)|32.8346|48.4275|0.3926|0.5185|
> |After|26.9926|57.7367|0.4889|0.5556|
>
> - **Concern about leakage from news updates (post-hoc corrections)**
>     - After collecting the raw data, we apply an LLM-based preprocessing pipeline to remove irrelevant or noisy content that is not aligned with the target time series.
>     - Human experts then manually inspect samples to verify that no future events or ex-post factual updates are leaked into the historical context (in contrast to the misaligned issue observed in Time-MMD).
>
> - **Details of the human evaluation protocol**
>     - During data construction, annotators explicitly check whether news articles or reports were updated after the target events, and whether such updates could inadvertently reveal future outcomes.
>     - Once the dataset is constructed, domain experts with at least a bachelor-level background in the relevant area evaluate each sample via a standardized form and:
>         - Verify that textual data are properly de-identified and do not expose specific entities that could be trivially memorized.
>         - Assess whether the historical analysis context is relevant to the past time series, and whether this context alone helps them predict the future direction; their predictions are then compared against ground truth labels.
>         - Check that the future outlook context is logically aligned with the change between the last historical point and the future target window.
>         - (By comparing Task 1 and Task 3) Examine the logical consistency of counterfactual contexts and whether they define coherent intervention-style scenarios.

---

> ### Author Response · Authors · 2025-11-28
> **Official Comment by Authors (Cont')**
>
> ### **5. Adding additional baselines and analyses of forecasting instances**
>
> - We are currently extending our set of baselines by including larger instruction-tuned LLMs such as Mixtral-8x22B-Instruct-v0.1 and Llama-3-70B-Instruct.
> - In the revised manuscript, we also plan to provide qualitative case studies of forecasting instances, illustrating how the models generate their predictions.
> - In particular, we will highlight examples where ambiguity in the textual context leads to unclear interpretations of the magnitude or scale of changes, and discuss how this interacts with our evaluation setup.
>
> ---
>
> ### **6. Strengthening the clarity and scope of the ablation studies**
>
> - Our existing ablations were designed to isolate the contribution of each modality by comparing:
>     - (i) time series only, (ii) time series + historical analysis context, (iii) time series + future outlook context, and (iv) time series + both historical analysis and future outlook context.
> - However, we acknowledge that the accompanying explanations did not clearly articulate the insights these comparisons were intended to convey.
> - In the revised manuscript, we will clarify the purpose and implications of these original ablations.
>
> - In addition, following the reviewer’s suggestion, we will introduce a new set of ablations that deliberately reduce the strength or explicitness of the future scenario text.
>     - For example, by removing the “likely to weaken significantly” part in the example of figure 1 to see if the model can deduce the impact of temperature on demand for natural gas itself.
> - These experiments will evaluate whether models remain robust and can still correctly utilize (or appropriately disregard) weaker or more indirect textual cues.
> - By combining the clarified original ablations with these newly introduced, reviewer-recommended settings, we aim to provide a more comprehensive and interpretable assessment of how different levels of contextual informativeness influence multimodal forecasting performance.
>
> ---
>
> ### **7. Planned reorganization of Section 4 for clearer explanation of the data construction pipeline**
>
> We plan to restructure Section 4 into four sequential subsections, each describing a distinct stage of the benchmark construction process.
>
> **(1) Data Collection**
>
> - Provide an overview of the time-series sources and their domain coverage.
> - Describe the textual sources used to construct historical analyses and future outlook scenarios.
> - Direct readers to the appendix for details specific to raw data resources.
>
> **(2) Preprocessing Pipeline (for Task 1 & Task 2)**
>
> - We will describe our three-stage LLM-based preprocessing procedure:
>     1. **Removing irrelevant or noisy content**
>         - LLM filtering followed by human verification to ensure that news articles are not updated after the fact and do not contain future events.
>     2. **De-identification to mitigate memorization leakage and to prevent overly specific cues**
>         - Show sample cases illustrating how sensitive or overly specific details were removed.
>         - Report that performance differences before/after the model’s knowledge cutoff boundary were negligible, further supporting that leakage was not occurring.
>     3. **Aligning the narrative with actual series changes**
>         - Explain how point-wise deltas (and their signs) were used to synchronize textual descriptions with the time-series movements.
>         - Clarify exceptions in domains where alignment behaves differently (e.g., energy and economy), with references to Appendix A.4.2.
>
> **(3) Multimodal Data Integration**
>
> - Distinguish between historical context and future outlook context and explain how they are combined with the time series to form multimodal samples.
> - For Task 3, describe how counterfactual contexts are derived from high-quality Task 1 samples by flipping the future outlook to generate internally consistent counterfactual scenarios (referencing Figure 1).
>
> **(4) Post-construction Validation**
>
> - **What we validate**
>     - Removal of noise and verification of proper de-identification.
>     - Relevance between historical time series and historical analysis context:
>         - Frozen-encoder representation similarity checks (TSFM and LLM),
>         - LLM-based relevance scoring,
>         - Human expert validation.
>     - Consistency between *history → future* transitions:
>         - Alignment between future outlook text and the actual future change interval.
>     - For Task 3, logical correctness of counterfactual scenarios
>         - Both LLM-based checks and human expert review.
>
> - **How validation is conducted**
>     - Stage 1: LLM-based automatic validation.
>     - Stage 2: Human expert evaluation protocol (double-checking), conducted via structured Google Forms with domain experts holding at least a bachelor’s degree in relevant fields.

---

### Official Review · Reviewer_WFsj · 2025-10-29

**Soundness:** 3
**Presentation:** 3
**Contribution:** 2
**Rating:** 2
**Confidence:** 3

**Summary:**

This paper introduces the WIT benchmark, a novel and valuable contribution to the field of multimodal time series forecasting.
Compared to existing benchmarks, this paper also construct the countrefactual task and incorporate more datasets to build a comprehensive benchmark. This paper's work try to bring multimodal time series forecasting to next level.

**Strengths:**

The strengths of this paper is the comprehensive dataset, collecting many real-world data which is not included in other multi-modal time series benchmarks.

**Weaknesses:**

1. Time Series Ground Truth in Counterfactual Forecasting (Task 3) is unclear. If text is constructed by mininal change, the time series should be changed too, but there is no clear description about how the time series can be change.

2. This paper mentioned some advantages of this benchmark over CiK. However, there is something good in CiK which is not in this benchmark, like: ** Retrieval and Reasoning **. CiK constructs multiple types of text to test different Retrieval and Reasoning abilities which is lacked in this paper. Also, this paper said CiK doesn't included the  counterfactual tasks, which is not true, because CiK contruct the text and the corresponding time series manually, so it must include some counterfatucal tasks, only not very categorized this way.
I think the real advantage of this paper over other benchmarks is the comprehensive real-world datasets instead of the counterfactual, because most of CiK data is not real in essence.

3. About the metrics. Besides the mse mae or some accuracy metrics, there should be some metrics to measure how the prediction obey the text. For example, there may be some cases with same MAE and MSE, but with totally different directions in the prediction.

**Questions:**

1. Counterfactual Ground Truth: For Task 3, what is the source of the ground truth time series used for evaluation after the textual context is modified? If the text is counterfactual, does the ground truth series remain the original (now misaligned) data, or is it also synthetically altered? This is unclear and critical for interpreting results.

2. Benchmark Scope vs. CiK: CiK explicitly tests retrieval and reasoning with mixed-context inputs, a capability WIT does not evaluate. What is the rationale for not incorporating such a test? Do you consider the lack of a retrieval/reasoning component a limitation of WIT's design?

3. Metrics for Textual Faithfulness: Beyond standard accuracy, how do you measure if a model's forecast directionally obeys the textual guidance (e.g., predicts "rise" when the text describes a "surge in demand")? Would a metric specifically for "textual guidance adherence" be more direct?

---

### Official Review · Reviewer_ALHr · 2025-10-29

**Soundness:** 2
**Presentation:** 2
**Contribution:** 3
**Rating:** 2
**Confidence:** 5

**Summary:**

The authors develop a benchmark for multimodal forecasting, with what-if scenarios and future events. The authors build this benchmark with real-world time series and real-world events that are paired together, and further with counterfactual events constructed with an LLM. The authors evaluate LLMs, multimodal time series models and unimodal time series models such as time series foundation models and statistical models, for their forecasting performance (MSE) on tasks from the benchmark and directional accuracy on counterfactual events.

**Strengths:**

* The idea of constructing counterfactual events and verifying directional accuracy (in the absence of the counterfactual time series) is very interesting. Applying this idea in the context of multimodal forecasting brings clear novelty.
* The de-identification rules that the authors apply on the data (mentioned in Sec 4.5) are well-thought out.
* As rightfully mentioned, instead of the approach of writing scenarios and modifying time series as in prior work, the approach of aligning real-world time series and events is interesting and brings the setup closer to real-world scenarios.
* The authors provide substantial detail on the data sources used for the benchmark and the prompts used for generating the final textual contexts in the benchmark. I appreciate this.

**Weaknesses:**

* Incorrect claims on related work:
  * The authors state “CiK [1] emphasizes contextual grounding and event understanding, but its design primarily supports retrospective reasoning rather than explicit, scenario-based forecasting.” This is incorrect. CiK [1] has tasks that are historical and future-based. In fact, there are tasks that are very similar to the tasks presented in this benchmark (see [this task](https://servicenow.github.io/context-is-key-forecasting/v0/UnemploymentCountyUsingMultipleStateData.html))
  * The authors state “These limitations motivate our benchmark: we provide expert-authored future scenarios that articulate plausible upcoming events”. This is exactly what CiK [1] did as well; the work however claims novelty on this aspect which is incorrect.

* Proposed static and dynamic types of context: This is already encapsulated in the 5 types of context that CiK [1] proposes (Intemporal, Historical, Future, Covariate, Causal information). The authors do not discuss the differences. It is not clear what value the proposed 2 types of context brings.

* Insufficient detail: No examples are provided of the tasks/instances in the benchmark. E.g. I do not know how fine-grained the contexts are. CiK [1] has overly specific contexts, but how is this benchmark different?

* Insufficient evaluation: The performance of LLMs without context is not provided in Table 3. So it is not clear if it is the context reasoning capabilities that is enabling the superior performance of the LLMs or if it is their forecasting capabilities. Further, it is not clear if they outperform the unimodal models when evaluated w/o context. This is important to know when LLMs are considered for use in forecasting in the real-world.
  * The performance of LLMs without any context for each domain is available in the appendix in C.2.2 (I appreciate that the authors put this) but the unimodal models are absent from this table. Further this is an important evaluation that must be visible from the main text without having to dig into the appendix.
  * The authors only evaluate LLMs with a prompting methodology similar to Direct Prompting as in prior work [1, 2, 3]. However this is not the only prompting methodology to use LLMs for forecasting; there also exists LLMP [4] / LLMTime [5] (prior work [1] evaluates the LLMP methodology). The authors must evaluate these strategies as well, at least with a subset of LLMs.

* Adding to the above point, the authors present a very limited analysis of results (Less than a single page when 5.2 and 5.3 are put together).
  *  CiK [1] showed that the largest LLMs (such as Llama-3.1-405B-Inst) perform the best, and that performance clearly differs with scale. Such analyses are not presented here.

  *  Further, a very limited set of LLMs are assessed. It is unclear why these LLMs were picked.

  *  The authors do not verify that the models obtain their performance due to the context reasoning capabilities and not their forecasting capabilities. This is related to the point about evaluating these models without context.

  *  The paper provides no insight on the limitations of LLMs and how they can be improved.

  *  And no examples are provided of the forecasts of LLMs with and without context, demonstrating how the context meaningfully changes the forecasts.

  *  An issue that the CiK paper [1] highlighted is the high cost of LLMs. That is not discussed here.


### Summary note

* My main reasons for rejection are the misleading claims of novelty, insufficient evaluation and lack of rigorousness in the paper. The benchmark appears to be rigorous but the paper doesn’t do a good job at convincing the reader, so I’m still on the fence [This is reflected in my scores for Soundness, Presentation and Contribution]
* My suggestion to the authors is to position their contributions accurately with respect to related work, and highlight precisely the novelty that they bring.

**Questions:**

* What is the difference between a conditional statement and anticipated event? That is not discussed in Sec. 3.1 and I don’t see the need to differentiate them. To the model it makes no difference in conditioning between either.
* The verification of the alignment and relevance of the text with the time series by human experts needs more explanation. This is crucial to the quality of the benchmark, as this is where prior works [6] have used different methodologies (such as an LLM judge).
  * How many annotators were there? What exact instructions were they given? What was the inter-rater agreement achieved?
  * Give examples of examples that were discarded after human verification.
* I completely miss the point of the ablation study 5.3. The authors state “In constructing the historical context for WIT, we extract all significant events corresponding to the history time series without any restrictions.” Why is this important? What is the relevance of this?
* A minor correction - “Yet, these models remain constrained by the quality of textual inputs, which in existing benchmarks are often descriptive or redundant, rather than predictive of future outcomes” → In this case isn’t their evaluation that remains constrained? The models themselves are not constrained.
* Minor / grammar: Figure 1 → “enables”: shoudn’t this be “enables evaluation of” ?

## References

[1] Williams, Andrew Robert, Arjun Ashok, Étienne Marcotte, Valentina Zantedeschi, Jithendaraa Subramanian, Roland Riachi, James Requeima et al. "Context is key: A benchmark for forecasting with essential textual information." ICML 2025.

[2] Ashok, Arjun, Andrew Robert Williams, Vincent Zhihao Zheng, Irina Rish, Nicolas Chapados, Étienne Marcotte, Valentina Zantedeschi, and Alexandre Drouin. "Beyond Na\" ive Prompting: Strategies for Improved Zero-shot Context-aided Forecasting with LLMs." arXiv preprint arXiv:2508.09904 (2025).

[3] Zhang, Xiyuan, Boran Han, Haoyang Fang, Abdul Fatir Ansari, Shuai Zhang, Danielle C. Maddix, Cuixiong Hu et al. "Does Multimodality Lead to Better Time Series Forecasting?." arXiv preprint arXiv:2506.21611 (2025).

[4] Requeima, James, John Bronskill, Dami Choi, Richard Turner, and David K. Duvenaud. "Llm processes: Numerical predictive distributions conditioned on natural language." Advances in Neural Information Processing Systems 37 (2024): 109609-109671.

[5] Gruver, Nate, Marc Finzi, Shikai Qiu, and Andrew G. Wilson. "Large language models are zero-shot time series forecasters." Advances in Neural Information Processing Systems 36 (2023): 19622-19635.

[6] Liu, Haoxin, Shangqing Xu, Zhiyuan Zhao, Lingkai Kong, Harshavardhan Prabhakar Kamarthi, Aditya Sasanur, Megha Sharma et al. "Time-mmd: Multi-domain multimodal dataset for time series analysis." Advances in Neural Information Processing Systems 37 (2024): 77888-77933.

---

### Official Review · Reviewer_LVWN · 2025-11-03

**Soundness:** 1
**Presentation:** 3
**Contribution:** 2
**Rating:** 2
**Confidence:** 4

**Summary:**

The paper presents "What if TSF (WIT)" benchmark to go beyond prediction to forecast under counterfactual scenarios. It combines expert crafted what if scenarios and future events with structured textual descriptions. For short term forecasting, MSE and trend accuracy (whether the trend will flip) is reported. For long horizon forecasting and in the counterfactual setting, just the direction of the trend is predicted and trend accuracy is reported based on whether the trend will flip or not. The benchmark contains 4 domains (politics, energy, society and economy) and the zero-shot performance of 6 models (Mistral 7B, Qwen2.5 7B, Qwen3-32B, GPT-4o and Gemma-3-27B, Time-MQA) are reported along.

**Strengths:**

- The paper proposes a multimodal counterfactual what-if benchmark. This has received little attention in the TSF literature, and a benchmark dedicated solely to evaluating this is both interesting and novel.

**Weaknesses:**

- **I am not sure if deidentification merely boils down to redacting company names and removing uniquely identifying URLs**. How can we be sure that memorization is not helping the model perform predictions after replacing <actual_company_name> with a placeholder like "COMP_A"? The news articles might still contain other identifying details. For example, "Amazon made deals with XYZ" can be anonymized to "COMP_A made deals...". However, it is not uncommon for news articles to contain details like "the tech giant's CEO, Jeff Bezos". Do the authors handle such cases? Or can they atleast provide empirical evidence from the prompts that go into the models?

- For table 3, the authors should also report MSE for long term and counterfactual tasks, as well as for the tables in the appendix.

- **The paper would benefit from added experiments on datasets released after cutoff dates of the LLMs used in the paper**. Even after the de-identification techniques used in the paper, it is possible the models might be able to pick up these closely related patterns from their pretraining data. However, this is not addressed by the authors. Given that this is a benchmarking paper, it is hard to comment on the rigor of the evaluations given an entirely LLM based pipeline. The gold standard to ensure there is no data leakage would be to evaluate on data released after the cutoff date.

- Future outlook context is generated by prompting GPT5-mini to list the single most significant sub event among summaries. Real life scenarios involve multiple complicated event-event interactions, sometimes even involving directly conflicting information, yet this is boiled down to listing one event for the prediction task.

- Other than the weaknesses already mentioned, the breadth of baselines (5 LLMs + Time-MQA) and tasks (4 domains) covered is also not particularly impressive for a benchmarking paper; the contributions are incremental/marginal. Even papers introducing methods (that also have to train model, rather than just evaluate zero-shot performance) typically evaluate on more baselines and tasks, despite not having benchmarking as the primary focus.

**Questions:**

- "To avoid confounding long term dynamics, counterfactual evaluation follows the short term forecasting setup of Task 1." – can the authors explain what they mean here?

- It is not clear what the ground truth values are for the forecasted variables in the counterfactual setting. How are these numbers obtained? Is it by prompting GPT5-mini? To my understanding both the counterfactual text and the counterfactual data is generated by using an LLM (GPT5-mini?). This generally seems very simplistic from the examples given in the paper. For example, in table 15, the future outlook text says: "Rising unemployment will sharply reduce consumer spending and push many households into financial distress.". The model then predicts "answer": {"future_timeseries": [69.92], "trend": "fall"}. This seems like neither a challenging task nor a realistic one? Can the authors comment on this?

---

### Note · Authors · 2025-12-05

I have read and agree with the venue's withdrawal policy on behalf of myself and my co-authors.